# Clinical cancer genomic profiling by three-platform sequencing of whole genome, whole exome and transcriptome

Michael Rusch [1,2], Joy Nakitandwe[2,3], Sheila Shurtleff[2,3], Scott Newman[1,2], Zhaojie Zhang[1,2], Michael N. Edmonson [1,2], Matthew Parker[4], Yuannian Jiao [1,2], Xiaotu Ma[1,2], Yanling Liu[1,2], Jiali Gu[2,3], Michael F. Walsh[2,5], Jared Becksfort[1,2], Andrew Thrasher[1,2], Yongjin Li[1,2], James McMurry[2,6], Erin Hedlund[1,2], Aman Patel [1,2], John Easton[1,2], Donald Yergeau[1,2], Bhavin Vadodaria[1,2], Ruth G. Tatevossian [2,3], Susana Raimondi[2,3], Dale Hedges[1,2], Xiang Chen [1,2], Kohei Hagiwara[1,2], Rose McGee [2,5], Giles W. Robinson[2,5], Jeffery M. Klco [2,3], Tanja A. Gruber[2,3,5], David W. Ellison[2,3], James R Downing[2,3] & Jinghui Zhang[1,2]

To evaluate the potential of an integrated clinical test to detect diverse classes of somatic and germline mutations relevant to pediatric oncology, we performed three-platform whole-genome (WGS), whole exome (WES) and transcriptome (RNA-Seq) sequencing of tumors and normal tissue from 78 pediatric cancer patients in a CLIA-certified, CAP-accredited laboratory. Our analysis pipeline achieves high accuracy by cross-validating variants between sequencing types, thereby removing the need for confirmatory testing, and facilitates comprehensive reporting in a clinically-relevant timeframe. Three-platform sequencing has a positive predictive value of 97–99, 99, and 91% for somatic SNVs, indels and structural variations, respectively, based on independent experimental verification of 15,225 variants. We report 240 pathogenic variants across all cases, including 84 of 86 known from previous diagnostic testing (98% sensitivity). Combined WES and RNA-Seq, the current standard for precision oncology, achieved only 78% sensitivity. These results emphasize the critical need for incorporating WGS in pediatric oncology testing.

[1] Department of Computational Biology, St. Jude Children's Research Hospital, 262 Danny Thomas Place, Memphis, TN 38105, USA. [2] Pediatric Cancer Genome Project, St. Jude Children's Research Hospital, 262 Danny Thomas Place, Memphis, TN 38105, USA. [3] Department of Pathology, St. Jude Children's Research Hospital, 262 Danny Thomas Place, Memphis, TN 38105, USA. [4] Sheffield Bioinformatics Core, University of Sheffield, 345a Glossop Road, Sheffield S10 2HQ, UK. [5] Department of Oncology, St. Jude Children's Research Hospital, 262 Danny Thomas Place, Memphis, TN 38105, USA. [6] Department of Information Services, St. Jude Children's Research Hospital, 262 Danny Thomas Place, Memphis, TN 38105, USA. These authors contributed equally: Michael Rusch, Joy Nakitandwe, Sheila Shurtleff, Scott Newman. Correspondence and requests for materials should be addressed to D.W.E. (email: david.ellison@stjude.org) or to J.R.D. (email: james.downing@stjude.org) or to J.Z. (email: jinghui.zhang@stjude.org)

Clinically important biomarkers for diagnosis, risk stratification, and targeted therapy for pediatric cancers include diverse types of somatic and germline genetic lesions[1]. For example, the diagnostic workup of pediatric leukemia alone evaluates the chromosomal ploidy (e.g. hyperdiploid for low risk, hypodiploid for high risk), gene fusions (e.g. BCR-ABL1 for high risk, ETV6-RUNX1 for low risk, PML-RARA for targeted therapy), complex re-arrangements (e.g. iAMP21 for high risk), copy-number alterations (CNA) or sequence mutations (e.g. IKZF1 disruption for poor prognosis), and other structural variations (e.g. FLT3 internal tandem duplication—ITD).

Traditional workflows to capture these diverse genetic abnormalities in the clinic are complex and multimodal, often including a combination of karyotyping, fluorescent in situ hybridization, copy-number microarray, quantitative RT-PCR, and Sanger sequencing. Such workflows lack the scalability and flexibility necessary to incorporate recently discovered lesions and are of limited use for studying cases with complex or non-standard findings[2–4].

The growing need to simultaneously interrogate a large number of loci within a clinically-relevant timeframe has directed a shift towards next-generation sequencing (NGS)[5]. Recently, clinical NGS tests using targeted gene panels or whole exome sequencing (WES) have been used to identify pathogenic sequence mutations including single nucleotide variations (SNVs) and small insertion-deletions (indels) in both adult[6–8] and pediatric cancer studies[8–10]. Several studies also incorporated

transcriptome sequencing (RNA-Seq) for detection of gene fusions and outlier expression[11,12]. Whole genome sequencing (WGS) is the most comprehensive platform for cancer genome profiling. However, clinical adoption of WGS has been limited to pilot studies involving very small numbers of cases (mostly < 10)[12–17] and the median turnaround time for generating the final clinical report may exceed 100 days due to mandatory secondary validation of variants by a CLIA-certified lab[12,18].

We hypothesized that a multi-platform NGS test combining WGS, WES, and RNA-Seq would improve both the comprehensiveness and the accuracy of detection and classification of somatic and germline variants important for cancer diagnosis, stratification, and treatment. Such testing could simultaneously simplify the diagnostic algorithm by replacing gene-specific or panel-based tests, remove the need for iterative testing and validation of the results by Sanger sequencing, and decrease the burden on those reviewing cases. To date, such a comprehensive approach remains largely unexplored, without any study to systematically evaluate the accuracy and diagnostic yield compared to alternative approaches.

We carried out a pilot study of three-platform sequencing by performing WGS and WES on paired tumor and normal samples and RNA-Seq on tumor samples in a CLIA-certified, CAP-accredited laboratory (Fig. 1a). Our selected 78 pediatric cancer patients had known biomarkers of diverse types including SNVs, indels, CNA and structural variants (SV) identified by multiple molecular pathology assays (Fig. 1b). We developed an analytical

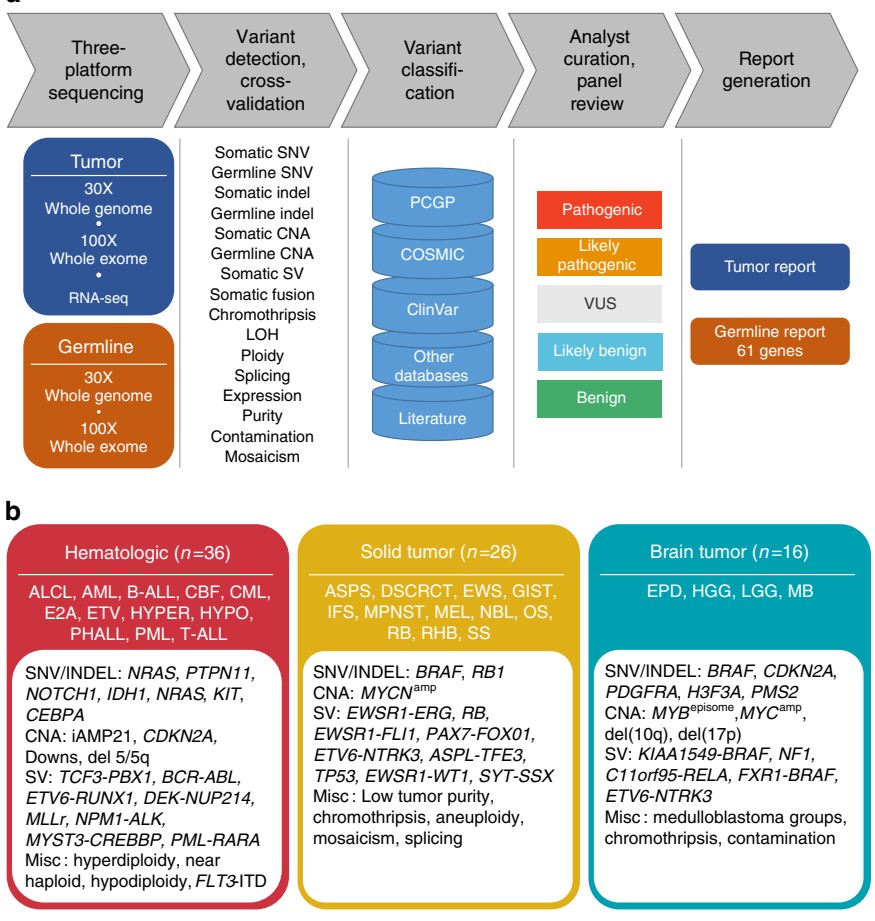

**Fig. 1** Design of clinical three-platform sequencing. **a** Overview of sequencing, variant detection, variant classification, panel review and report generation. Chromothripsis was used as an annotation for ploidy report and we followed guidelines by Korbel and Campbell[59]. **b** Selection of 78 pediatric cancer patients with biomarkers identified by multiple molecular pathology assays

pipeline to detect, integrate and cross-validate somatic[19,20] and germline variants predicted by each platform and to perform preliminary pathogenicity classification that was then reviewed by a multidisciplinary expert panel. We performed extensive experimental validation to establish the sensitivity and positive predictive value (PPV) of three-platform sequencing, and assessed the diagnostic yield of clinically important somatic and germline variants of each platform either alone or in combination.

We show that three-platform sequencing achieves both high sensitivity of 98% of known pathogenic variants and a high PPV of 97–99, 99, and 91% for somatic SNVs, indels and structural variations, respectively. In contrast, 78% of the pathogenic variants were detected by combined analysis of WES and RNA-Seq, which represents the current gold standard for precision oncology. The results of our study emphasize the critical need for incorporation of WGS in clinical sequencing, particularly in the context of pediatric oncology.

## Results

**Case selection.** Seventy-eight pediatric cancer patients, including 36 with hematologic malignancies, 16 with brain tumors, and 26 with other solid tumors, were selected for three-platform sequencing (Supplementary Data 1). These cases harbored diverse types of clinically significant genomic alterations that had been detected previously by a variety of molecular assays (Fig. 1b; Supplementary Data 2). The cases were chosen to be broadly representative of the pediatric cancers treated at our institution, but also to be genetically heterogeneous with differing tumor purity, ploidy, structural genome complexity, and mutational burden (Supplementary Data 2 and 3). We specifically included genetic lesions and tumor specimens that are challenging to detect or interpret by NGS such as *FLT3*-ITD, *KIAA1549-BRAF* fusion, variants from high-GC exons[21], complex structural rearrangements, and samples with low tumor purity, intra-tumor heterogeneity, and/or tumor-in-normal contamination (Supplementary Data 2, 3). The average tumor purity was 0.81 (range 0.21–1.00) and 14% (11/78) of the tumor specimens had purity < 0.5 (Supplementary Data 3).

Research WGS had been performed previously for 33 tumor-normal sample pairs as part of the St. Jude Children's Research Hospital/Washington University Pediatric Cancer Genome Project (PCGP). The average coverage of PCGP WGS for tumor and normal samples ranged from 29X to 83X and 24X to 43X, respectively, as low purity tumor specimens were sequenced at high depth (Supplementary Data 3). Prior findings made from PCGP provided an important benchmark for validating our sequencing and analytical processes and for assessing the sensitivity of three-platform sequencing. For the present study, new clinical WGS, WES, and RNA-Seq data were generated for all 33 overlapping cases. DNA for 13 tumors and 17 normal samples were re-extracted from a different isolate while the same specimens were used for the remaining 20 tumors and 16 normal samples.

**Sequence coverage.** DNA and RNA were extracted from frozen tumor tissue while matched normal DNA was extracted from peripheral blood, remission bone marrow, histologically normal tissue, and sorted T-cells (Supplementary Data 1). Crucially, we used a PCR-free whole genome sequencing strategy that minimized amplification-related coverage bias and sequencing artifacts (Supplementary Fig. 1). We achieved a mean coverage for tumor and normal WGS of 38X and 36X respectively, and 110X and 103X for tumor and normal WES respectively (Supplementary Fig. 2; Supplementary Data 3). RNA-Seq had ≥ 20X coverage of 30% of exons for tumor total RNA (Supplementary Data 4).

90% of the germline WGS and 90% of the germline WES samples met a target of ≥ 80% coding exons with ≥ 20X coverage required for high-quality variant detection (Supplementary Data 4).

**Analysis overview.** We first analyzed sequence data with an automated pipeline developed specifically for three-platform sequencing. The pipeline detected both somatic and germline SNVs, indels and ITDs, arm-level and focal CNAs and loss-of-heterozygosity (LOH) as well as somatic SVs and RNA-Seq gene fusions. It also estimated tumor purity using variant allele fraction of polymorphic germline SNVs within CNA and LOH regions. Central to our automated framework was cross-validation, where evidence from two or three-platforms was combined to support the validity of each variant. Preliminary variant pathogenicity classification was carried out by automated searching of multiple somatic and germline mutation databases and in silico prediction of the functional impact of each variant.

Following the automated pipeline run, an analyst manually reviewed high quality or cross-validated variants, flagging those with potential clinical relevance. This included all types of somatic variation genome-wide, as well as germline non-silent mutations and CNAs affecting 61 cancer predisposition genes[22] (Supplementary Data 5). Where pertinent, mRNA expression, aberrant splicing, and germline mosaicism were also investigated. Lastly, the analyst assessed the existence of tumor-in-normal contamination—for example, when leukemic blasts are found in peripheral blood—by examining the mutant allele fraction (MAF) of likely somatic mutations in the germline WGS and WES samples. A multidisciplinary panel of experts then reviewed each potentially clinically-relevant variant, taking into account all available cross-platform information. Separate draft reports were generated for committee-approved somatic and germline pathogenic and likely pathogenic variants.

**Integrative analysis of genomic alterations.** An important aspect of our analysis was the cross-platform integration of genomic alterations. For SNVs and indels, this proved extremely useful in eliminating false positive predictions (Supplementary Note 1). The specific details of SNV and indel cross-platform validation are included in the Methods section while our integrated SV/CNA analysis is discussed below.

Structural and copy number abnormalities are key biomarkers for pediatric oncology, and integrated SV/CNA/gene fusion analysis using three-platform sequencing significantly improved detection over any single platform. We detected somatic SVs from WGS using CREST[23], and performed integrative SV/CNA detection using CONSERTING[24] to identify subclonal or complex SVs linked to CNAs. For comparison purposes, we also performed CNA analysis on WES using the Sequenza algorithm[25]. We detected RNA-Seq gene fusions, ITDs and other disruptions arising from truncations, promoter swaps and readthrough using Cicero (Li et al., in preparation). SVs/CNAs derived from DNA were merged with matching RNA-Seq events, allowing the analyst to review the variant and evaluate its functional impact with a view of all available evidence (Fig. 2a).

An example of integrative DNA–RNA SV analysis is demonstrated in the detection of a complex *PDGFRA* gene fusion in a high-grade glioma (Fig. 2b). This sample contained a high-level complex amplification that included exons 10–23 of *PDGFRA* with exons 1–9 being excluded from the amplified region. Three fusion transcripts were detected in RNA-Seq, connecting exon 1 of *DIP2C* located on chromosome 10 to each of exons 10, 11, and 12 of *PDGFRA* on chromosome 4. A DNA SV detected in WGS has one breakpoint in *PDGFRA* intron 10 and the other in *DIP2C* intron 1. In RNA-Seq, two fusion transcripts involving *PDGFRA*

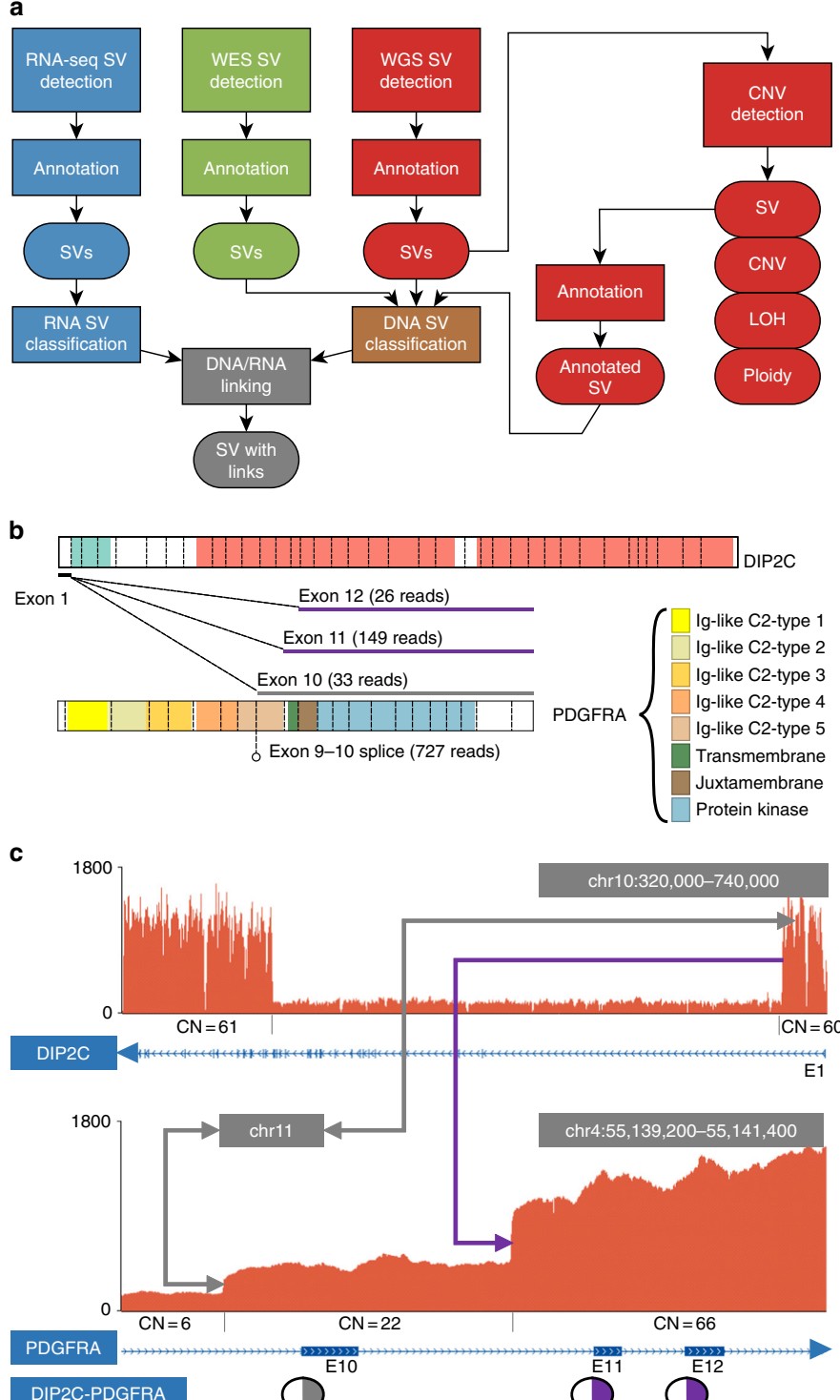

**Fig. 2** Integrative analysis of structural variation. **a** Multi-platform, multi-analysis approach for detecting SV by three-platform sequencing. SV detection is run using RNA-Seq and WES of the tumor sample, and integrated SV and CNA detection are run using WGS of the tumor and normal samples. **b**, **c** An example of linking three fusion transcripts of *DIP2C-PDGFRA* detected in RNA-Seq to DNA SVs detected in WGS in a high-grade glioma. The fusion transcripts are shown in protein view (**b**) with the domains marked in color and the vertical dotted lines marking the boundaries of each exon with chimeric RNA read counts indicated above the *PDGFRA* ideogram and wildtype exon 9–10 read counts below. The fusion transcript involving *PDGFRA* exon 10— marked by the gray line in the *PDGFRA* protein view (**b**)—is linked to a novel DNA SV involving chromosome 4, 11 and 10, shown using the gray line in (**c**), while the other two fusion transcripts, marked by the purple lines in (**b**), are linked to the same DNA SV involving chromosome 4 and 10, shown using the purple line in (**c**)

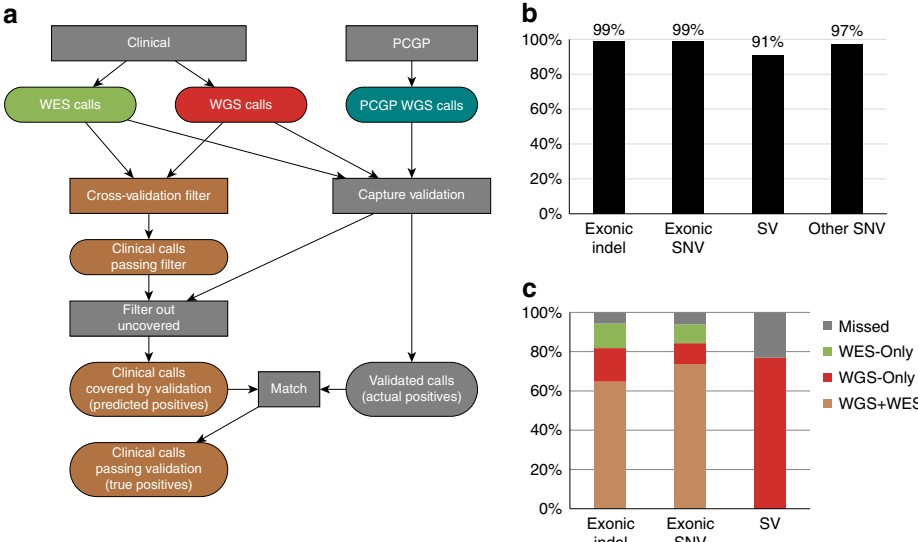

**Fig. 3** Accuracy of somatic variant detection based on capture validation of 18 cases with PCGP data. **a** Design of capture validation for measuring sensitivity and PPV of our analytical pipeline using somatic exonic SNV/indel detection as an example. For exonic SNV/indel, variants that passed the cross-validation filter and had adequate coverage from custom capture sequencing were considered "predicted positives". Predicted positives that were validated by capture sequencing are considered true positives whereas all variants—including those that failed to be detected or those filtered by cross-validation filtering—that were validated by capture sequencing are considered "actual positives". **b** Summary of PPV (true positives/predicted positives) for each variant type. The predicted positive variants for exonic indels and SNVs are the variants that pass the cross-validation filter, whereas other SNVs, which refers to high-confidence non-exonic SNVs, are based on WGS only; as such, they are reported separately. Our test does not report non-exonic indels, so they are omitted. **c** Summary of sensitivity (true positives/actual positives) for each variant type. For exonic SNV/indel, most of the variants are detected by our variant detection pipeline on both WGS and WES (WGS+WES) platforms; of those that are detected by our pipeline on one platform, results for WGS and WES were comparable, with slightly more detected by WGS. The "Missed" variants are the false negatives; they were removed by cross-validation filtering or only detected by PCGP

exons 11 and 12 can be mapped to this DNA SV. However, the fusion transcript involving *PDGFRA* exon 10 is mapped to a different DNA SV that connects *PDGFRA* and *DIP2C* via an intermediate segment on chromosome 11. This 3-chromosome SV matched the boundary of a CNA with an estimated copy number of 22 on *PDGFRA* and its connection to the *PDFGRA* fusion was not recognized previously in our research profiling of this tumor[26]. The presence of multiple SVs, CNAs and fusion transcripts suggests that *DIP2C-PDGFRA* re-arrangement may have occurred independently either in the same tumor cell or in different subclones and that the PDGFRA fusion resulted in a constitutively active kinase[27,28] (see Supplementary Note 2 for more details).

**Analytical performance of somatic variant detection**. We next evaluated the performance of somatic variant detection by three-platform sequencing. We focused our analysis on 18 cases that had been sequenced in the PCGP and had validation sequencing by custom capture. We used the variants that were successfully captured during the validation experiment to evaluate sensitivity and PPV.

To determine positives for calculating PPV, we used final variant calls, which were the product of quality filtering, cross-validation analysis to integrate variant calls from multiple platforms (Methods), and manual review for all mutation types except non-exonic SNVs, for which we included only high confidence variants (Fig. 3a, Supplementary Data 6A–D). Three-platform sequencing achieved high PPVs of 99% for 662 exonic SNVs, 99% for 84 exonic indels, 97% for 12,978 other non-exonic SNVs and 91% for 1493 SVs (Fig. 3b). We extended the analysis to an additional 20 cases with capture validation but no PCGP data, and observed similar validation rates of 98, 99, 98, and 86%,

respectively (additional data is made available online at https://pecan.stjude.cloud/permalink/cts).

To evaluate sensitivity, we assembled a truth dataset composed of validated variants detected by various molecular pathology methods, PCGP and three-platform sequencing. Previously, research WGS of matched tumor and normal samples was performed at the Washington University Genome Sequencing Center (WUGSC), and somatic variants were predicted by two independent bioinformatics pipelines (St. Jude and WUGSC) and validated by custom capture sequencing[21]. Structural variation prediction was unsuccessful in the PCGP for the case SJHGG001 due to poor WGS quality, so this case was omitted from the sensitivity calculation for SVs. Combining the validated variants resulted in 695 exonic SNVs, 88 exonic indels and 1759 SVs. Three-platform sequencing achieved a high sensitivity by detecting 94% of SNVs, 94% of indels and 76% of SVs (Fig. 3c).

As shown in Fig. 4, a single osteosarcoma, SJOS013, contributed 25% of the false negative exonic SNVs (10/42) and 75% of the false negative SVs (310/415) in our cohort. The unusually low sensitivity was likely caused by sample character-istics as tumor DNA from SJOS013 was re-extracted from a different isolate. SNVs found by both studies exhibited a 50% reduction in MAF in the clinical pilot sample relative to the PCGP sample (Fig. 4b). By contrast, the corresponding relative MAF values for an unrelated osteosarcoma (SJOS001) were in near perfect agreement (Fig. 4a). The reduced sensitivity is, therefore, most likely due to tumor heterogeneity and, specifically, lower tumor purity of the sample analyzed by the clinical pilot study. Another sample with low sensitivity in SNV detection (79%) was SJHGG001, in which the germline sample was taken at limited autopsy from the cerebellum relatively close to the pontine tumor. Although infiltrating tumor cells were not

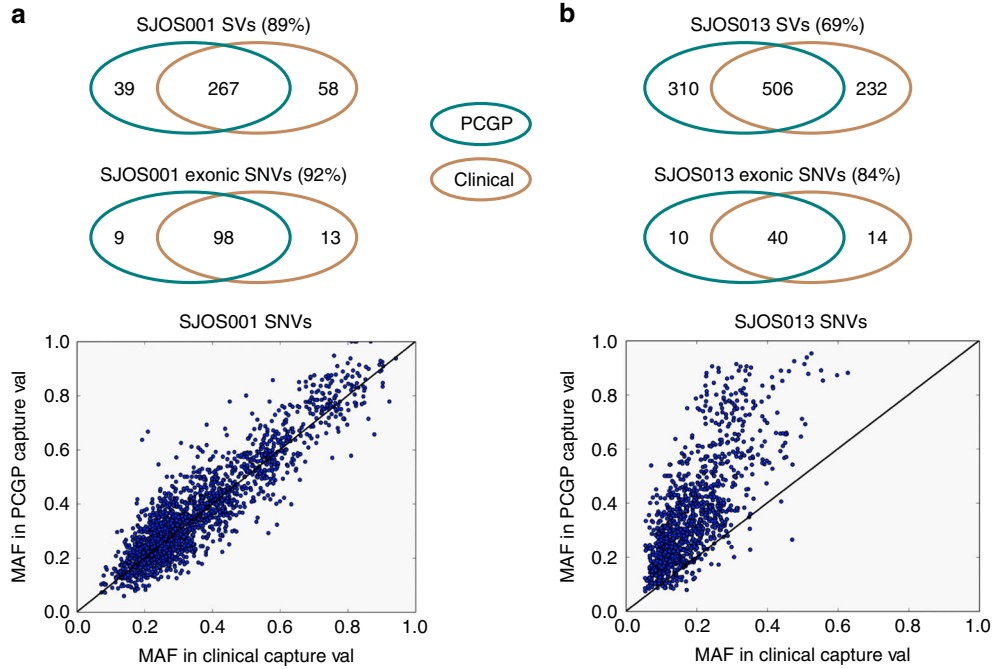

**Fig. 4** Comparison of clinical-PCGP concordance of two osteosarcoma cases. For each case, two venn diagrams show overlap between PCGP calls and clinical calls, and a scatter plot compares the SNV MAF obtained by capture validation performed during the present clinical study (x-axis) and previously by the PCGP validation lab (y-axis). The SNVs used for a MAF plot included both exonic and non-exonic variants. The tumor samples used for both cases were different than the tumor samples in PCGP, and both showed lower-than-average sensitivity. **a** MAFs from SJOS001 SNV calls in PCGP and Clinical were highly correlated, and the SVs and exonic SNVs showed higher correlation. **b** The lower tumor purity of the SJOS013 clinical sample is reflected in the MAF distribution. Differences between the PCGP and clinical samples in this case reduced the measured sensitivity

identified microscopically, we estimated 25% tumor contamination. Surprisingly the false negative variants were subclonal in the tumor and had comparable MAFs in tumor and normal. This suggests that the infiltrating tumor cells contaminating the normal tissue were dominated by a minor subclone in the bulk tumor (Supplementary Fig. 3).

Finally, we evaluated limits of detection using a statistical model and in silico resampling. The analysis showed a > 95% probability of detecting variants with a MAF of 0.1 (Supplementary Fig. 4, Supplementary Methods).

**Detection of diverse types of pathogenic variants**. We next assessed the diagnostic yield of three-platform sequencing for all 78 cases (Fig. 5a). Our expert panel reviewed a median of 38 variants per case assigning "Pathogenic" (P) or "Likely Pathogenic" (LP) status to 240 variants (229 somatic and 11 germline) with an average of 3.1 P/LP findings per sample. The P/LP somatic variants included 54 SNVs, 29 indels, 7 ITDs, 44 gene-fusing SVs and T-cell receptor translocations, 59 gene-disrupting SVs/focal CNAs and amplifications, 33 arm or chromosome level CNAs and 3 LOH events. The P/LP germline variants included eight SNVs, one indel, and two CNAs. Eighty-six P/LP variants had previously been detected either by standard molecular pathology and cytogenetics (n = 55) or exclusively as part of the PCGP (n = 31) (Supplementary Data 7A).

Despite less than 50% tumor purity in 11/78 cases (14%) and nine cases with > 5% tumor-in-normal contamination in matching germline samples, three-platform sequencing detected all abnormalities (55/55) found by standard molecular pathology testing and all but two abnormalities (29/31) found additionally by PCGP. The missing events were a high-level episomal MYCN amplification in retinoblastoma SJRB051 and an FXR1–BRAF fusion in low-grade glioma SJLGG026. These omissions were

likely due to heterogeneity between the PCGP samples and those used for this study, and low variant frequency in the tumor, respectively (see Supplementary Notes 3, 4). SJRB051 had a complex genome structure (due to chromothripsis) that included biallelic rearrangement of RB1 on chromosome 13 (Supplementary Fig. 5) with an estimated tumor purity of 0.75 and 0.97 for the PCGP and clinical pilot sample, respectively (Supplementary Data 3). While the CNAs and SVs on chromosome 13 were in perfect agreement with PCGP results, the MYCN locus on chromosome 2 lacked any read-depth increase or SV-supporting reads which were prominent in the PCGP data (Fig. 5b and Supplementary Note 3). The FXR1–BRAF fusion was originally detected with a MAF of approximately 0.05 in 65X WGS generated by the PCGP[29]. Manual review of 37X WGS in this study uncovered only two SV-supporting reads, insufficient for the automated pipeline to call the variant (see Supplementary Note 4).

Germline results included one case of trisomy 21, consistent with the previous diagnosis of Down Syndrome, one intragenic deletion of TP53 (medulloblastoma, discussed below), TP53 R280S (hypodiploid ALL), TP53 N235S (MPNST), mosaic IDH1 R132H (AML), truncations in PMS2 (high grade glioma), NF1 (low-grade glioma), APC (medulloblastoma), mosaic RB1 (retinoblastoma), BRCA2 (rhabdomyosarcoma), and SDHA (GIST). In most cases, a second somatic hit was evident in the tumor sequence.

All but two P/LP SNV/indels (90/92) had cross-platform WGS/WES support. The two variants exclusive to WGS, SH2B3 R216fs, and FLT3 D835Y, were poorly covered in WES across all samples (mean WES coverage was 0 with a range of 0–3.2X; Supplementary Fig. 6, Supplementary Note 5). An additional five variants were detected by WGS alone but had supporting reads in WES, but too few for de novo detection; details are provided in

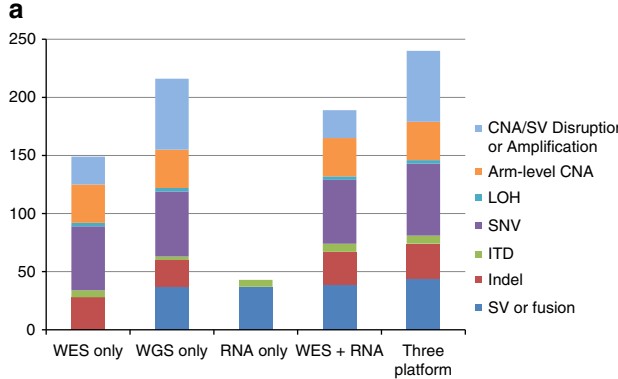

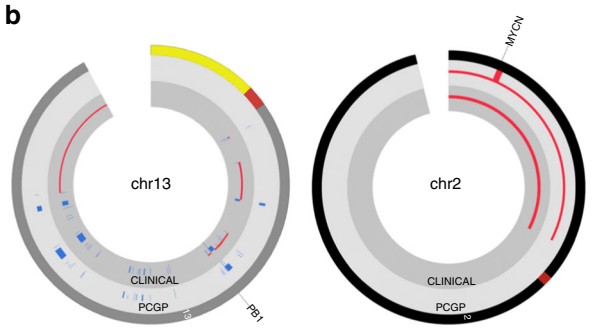

**Fig. 5** Detection of pathogenic or likely pathogenic variants by three-platform sequencing. **a** Number of variants detected by each platform either alone or in combination. **b** Comparison of copy number alterations detected by PCGP (outer circle) and three-platform sequencing (inner circle). The left CIRCOS plot depicts the CNAs detected on chromosome 13 showing near-perfect agreement between the samples analyzed by PCGP and three-platform sequencing. The right CIRCOS plot depicts chromosome 2, where a *MYCN* amplicon was detected only in the PCGP sample

the Supplementary Methods and Supplementary Data 7C. We also noted that 13 P/LP SNV/indels with MAF ranging from 0.01 to 0.17 were detected by WES alone but manually cross-validated by WGS. The majority (10/13) of these variants had MAF below 0.1, indicating higher WGS coverage may increase the power for detecting these subclonal variants (Supplementary Data 7B).

Structural variants (including ITDs) accounted for 46% (110/ 240) of P/LP variants and benefited the most from the multi-platform approach. ITDs ranged from 24 to 96 bp, and were detected by several different combinations of WGS, WES and/or RNA-Seq with WES performing the best overall finding 6/7. For gene-fusing SVs, 83% (35/42) had two-platform support while 17% (7/42) were supported by only one platform. Specifically, WGS enabled detection of fusions with low expression (e.g. *KMT2A-MLLT3*, *ETV6-RUNX1*, and *KIAA1549-BRAF*)—which we confirmed by inspecting RNA-Seq expression values of partner genes (Supplementary Data 7A) and by performing RT-PCR (Supplementary Note 6, Supplementary Table 1)—whereas RNA-Seq recovered potentially repeat-associated and complex rearrangements difficult to detect or interpret using genomic DNA alone (e.g. *RUNX1-RUNX1T1*, *BCL11A-GRIP2*, *FYCO1-RAF1*, and *KMT2A-AFF1*). Thus, relying on RNA alone would have caused us to miss 3/42 gene fusions (7%). Additionally, WGS allowed us to unambiguously identify two T-cell receptor rearrangements, *TCR-LMO2*, with no RNA junction support but the high expression and *TCR-NKX2-1* with an RNA junction 1.3 kb downstream of the target gene.

**Diagnostic yield of two versus three-platform sequencing.** The utility of three-platform sequencing is highlighted when we consider the yield of all P and LP variants found by each platform independently: RNA-Seq detected 18% (43/242), WES (including WES-derived CNAs) detected 62% (149/242) and WGS had the highest standalone platform value, detecting 89% (216/242). Three-platform sequencing, detected 99% of variants (240/242).

We compared three-platform sequencing with the current gold standard of combined WES and RNA-Seq. For the purposes of this study, we evaluated the approach of combined WES and RNA-Seq on a research basis rather than as a full clinical assay validation. Combined WES and RNA-Seq identified 78% (188/ 242) of P/LP variants. The lower yield relative to three-platform sequencing was mostly driven by focal and exon-poor CNVs ($n =$ 36) that were not detected in our exome-only copy number analysis (Supplementary Fig. 7, Supplementary Data 8). Notable missed alterations included focal and intragenic disruptions of *CDKN2A*, *CREBBP*, *ETV6*, *MYB*, *NF1*, *NR3C2*, *PAX5*, *PTEN*, *RB1*, and *TP53* (Supplementary Data 7A and Supplementary Fig. 7).

**Novel findings due to the inclusion of WGS.** In addition to robust detection of P/LP variants, three-platform sequencing provided biological insight into our patient samples. For example, the acute myeloid leukemia SJAML030006 had a *DEK-NUP214* fusion. Manual review found the same SV junction in approximately 4% of the matched normal WGS reads—indicating 8% tumor in normal contamination (Fig. 6a, Table 1). This was not surprising as the normal sample, acquired one month after diagnosis, was known to contain 10–15% blasts. The same tumor had a *FLT3*-ITD but we did not detect any SV junctions in the matched normal in spite of the ITD having a higher tumor MAF (due to chromosome 13 LOH) than the fusion. Statistical modeling of variant frequencies and read depth showed that the discrepancy was unlikely to have occurred by chance ($p = 0.00046$ by binomial test). This is consistent with a model in which *DEK-NUP214* was an early genomic alteration, present in all tumor cells at diagnosis, while the *FLT3*-ITD was acquired later and present in approximately 80% of the tumor. The *FLT3*-ITD subclone, which also contained the *DEK-NUP214* alteration, was eradicated/diminished by therapy, while those with only *DEK-NUP214* remained (Fig. 6a).

A second example illustrates the detection of a germline mutation by three-platform sequencing that may impact therapy. SJMB030020 was classified as a medulloblastoma with sonic hedgehog (SHH) pathway activation by immunohistochemistry, a standard clinical assay[30], but lacked any canonical somatic mutations in the SHH pathway. Three-platform sequencing revealed a biallelic loss of *TP53*: the first hit came from a 22 kb deletion of exons 1–9 in the germline, consistent with a new diagnosis of Li Fraumeni syndrome, and the second from a 13.7 Mb somatic deletion of chromosome arm 17p (Fig. 6b). This focal deletion was below the resolution of WES CNA analysis and was only detected by WGS. The tumor had a highly rearranged genome with high-level focal amplifications of *MYCN* (319 CN; 660 FPKM), *CCND2* (123 CN; FPKM 2516) and *GLI2* (11 CN; FPKM 24). Co-amplification of *MYCN* and *GLI2* have been described previously in SHH medulloblastoma[30–32] and, in the context of *TP53* pathway defects, is associated with a very poor outcome[33,34]. Following WHO guidelines, the combined findings allowed us to classify the tumor as a medulloblastoma, SHH-activated, *TP53*-mutant. However, the amplification and extremely high expression of *CCND2* (among the highest in >900 PCGP tumors) suggested that the RB pathway was also compromised in this tumor. This amplification may have

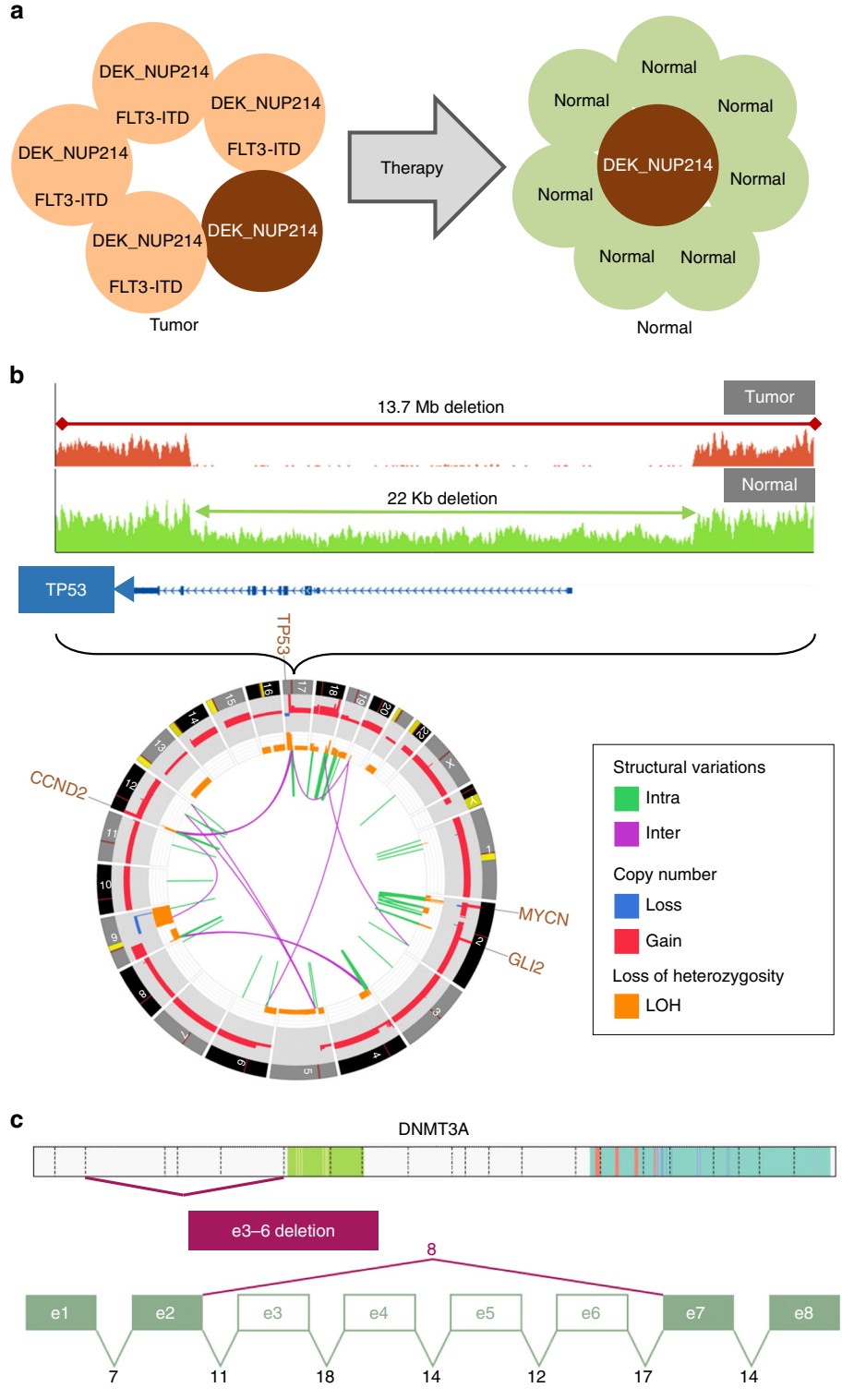

**Fig. 6** Examples of novel findings due to inclusion of WGS. **a** Illustration of residual tumor detected in the normal sample of an AML. Associated read counts can be found in Table 1. **b** Germline *TP53* deletion detected in a medulloblastoma. The tumor genome has a second somatic hit (i.e. a 13.7 Mb deletion) resulting in homozygous deletion of *TP53*. The tumor genome, shown in CIRCOS, underwent complex re-arrangement resulting in high amplification in *GLI2*, *MYCN*, and *CCND2*. **c** A variant of uncertain significance (VUS) in *DNMT3A* in a neuroblastoma. This variant deletes exons 3–6 of *DNMT3A* with aberrant transcription predicted to result in an in-frame deletion of the protein. The purple line linking exons 2–7 is annotated with the RNA-Seq read count of the aberrant splice junction caused by the deletion. The read count supporting each canonical splice junction is displayed underneath

**Table 1 Variant frequencies in AML with residual tumor in the germline sample**

| Variant | Mutant reads in tumor | Total reads in tumor | MAF in tumor | % Cells in tumor | Mutant reads in germline | Total reads in germline | MAF in germline | % Cells in germline |
|---|---|---|---|---|---|---|---|---|
| DEK_NUP214 | 15 | 34 | 0.44 | 88% | 4 | 103 | 0.04 | 7.7% |
| FLT3 ITD | 96 | 120 | 0.8 | 80% | 0 | 121 | 0 | 0% |

potential therapeutic implications as tumors overexpressing Cyclin D family members can respond to *CDK4/6* inhibition[35,36].

The final examples demonstrate novel variants detected by WGS that would be difficult to detect by WES or RNA-Seq. The first was a 50 kb somatic deletion detected by WGS CNA analysis but not by WES within *DNMT3A* in neuroblastoma SJNBL030014. A novel in-frame transcript lacking exons 3–6 was detected in RNA-Seq (Fig. 6c). Aberrant splicing was present in 30% of the transcripts and in the absence of supporting evidence from WGS would have simply been considered an alternatively spliced isoform. Although *DNMT3A* is a known driver gene in AML[37], the significance of this finding in neuroblastoma is uncertain at present.

Similarly, WGS detected a 96 bp *NOTCH1*-ITD in T-ALL, SJMLL002. This ITD was atypical as it encompassed 70 bp of intron 27 and the first 26 bp exon 28. As the ITD extended into an intron, the region had very low coverage by WES and no ITD was detected by this platform. Cross-validation by RNA-Seq and in silico translation showed that the ITD preserved the reading frame (Supplementary Fig. 8); we hypothesize it was functionally equivalent to recurrent *NOTCH1* juxtamembrane region ITDs in T-ALL[38].

## Discussion

A multidisciplinary clinical genomics working group was established at St. Jude Children's Research Hospital to develop and validate three-platform sequencing for molecular diagnosis of pediatric oncology patients. We assessed the test's PPV and sensitivity according to the guidelines by both CAP[39] and the Next Generation Sequencing-Standardization of Clinical Testing working groups[40]. We demonstrated experimentally that three-platform sequencing had near perfect PPV and sensitivity, producing a high yield of clinically relevant P/LP variants.

The three-platform sequencing approach provided a unique opportunity to assess the value of adding WGS to other genome-wide platforms more commonly used for clinical tumor profiling. We showed that, independently, WGS detected 89% of P/LP variants compared to only 62% by WES and 18% by RNA-Seq. When used in combination, three-platform sequencing detected 99% of P/LP variants compared to only 78% using combined WES and RNA-Seq. On a per subject basis, 36/78 (46%) had one or more additional P/LP variants detected by the three-platform test when compared to combined WES and RNA-Seq. We consider these disparities important to highlight in an era of precision oncology.

Three-platform sequencing confers several advantages over single-platform NGS and alternative multi-platform approaches such as a targeted NGS gene panel, exome plus RNA-Seq or copy number microarray. First, it allows robust detection of variants that a single NGS platform might miss or prove difficult to interpret, such as repeat-mediated/complex translocations, gene fusions with low expression and focal CNAs.

Second, it easily facilitates clinical reporting of newly discovered driver genes, for example, *IGH-DUX4* and *IGH-EPOR* rearrangements in B-ALL[41,42]. By contrast, adding new genes to a panel-based test requires laborious cycles of re-design, re-

sequencing, and re-validation, as well as additional patient tissue or nucleic acids.

Third, the high accuracy achieved by cross-platform validation obviates the need for secondary validation (Fig. 3, Supplementary Note 7). Our validation capture experiment showed that an SNV or indel can be considered validated if it can be detected independently in both WGS and WES. Likewise, if a focal CNA or RNA-Seq fusion is supported by a WGS SV, it is virtually certain to be a true genomic alteration. Our data show that, paradoxically, combined tumor/normal WGS/WES/RNA-Seq analysis produces a similar number of reviewable mutations (a median of 38 variants per case) as a small, 50–100 gene, unpaired NGS panel[43]. This simultaneously removes the need for iterative testing and decreases the burden on those reviewing and signing out cases. These related innovations have decreased turnaround time to the point where real-time clinical testing can happen. Generation of a curated mutation report from raw sequencing data was estimated to take 8 days based on batch analysis of the samples used in the study. Allowing for library preparation and sequencing time, we extrapolated a clinical turnaround time of 28–42 days. We have since deployed three-platform sequencing as a clinical service and achieved a median turnaround time of 31 days from sample receipt to report, which is comparable to a median turnaround time of 3 weeks reported in two recent clinical genomic studies that employed tumor exome sequencing[44] or integrated panel/selected transcriptome sequencing[45].

Fourth, integration of WGS and RNA-Seq enhances our power to classify SVs and CNAs. RNA-Seq can, for example, show disruption of the reading frame caused by an intragenic CNA (e.g. *NF1* in SJHYPO120). Outlier RNA-Seq expression can also help us discover the likely drivers within complex chromothripsis rearrangements or co-amplifications (e.g. *MYCN*, *GLI2*, and *CCND2* in SJMB030020) and assess the functional impact of immunoglobulin and T-cell receptor translocations (e.g. *TCR-LMO2* in SJTALL077). Without this corroborating evidence, the functional impact of such events can merely be considered predictions, making their interpretation difficult[46].

Lastly, three-platform sequencing can serve as a discovery engine. Despite efforts by the PCGP and other pediatric cancer genomics initiatives, many uncommon pediatric tumors require improved genomic characterization, and three-platform sequencing generates candidates for functional follow-up (e.g. *DNMT3A* in neuroblastoma). Further, the incorporation of WGS could, in the future, allow for assessment of telomere length[47,48], detection of viral integration[49], and discovery of driver mutations in non-coding regions[50,51].

The disadvantages of three-platform sequencing are the computing infrastructure required for data storage and analysis, and the cost of sequencing. Data retained from each paired patient sample occupies 150–200 Gb and, in line with our clinical laboratory protocols, must be rapidly accessible for 6 months which is estimated to cost $25 per case in a cloud computing platform. For this pilot study, sequencing was carried out using Illumina HiSeq 2000/2500 machines at a cost of $25,000/case ($18,000 for 30x WGS). However, costs have since been reduced to $8,600/case ($6,600 for 45x WGS) in our clinical service as samples are now sequenced on HiSeq 4000 machines. The

increase in WGS coverage from 30x to 45x in clinical service, when integrated with 100x WES data, is expected to improve the power for detecting variants with variant allele frequency $\geq 0.1$ from 95.7 to 99.1% (Supplementary Fig. 4). We anticipate that technology development will further reduce the cost for both computing and sequencing. As treatment of pediatric cancer typically costs hundreds of thousands of dollars, the precision of integrative genomic profiling offered by three-platform sequencing outweighs the extended time for secondary validation and the potential for missing ~ 20% pathogenic variants in pediatric cancer.

In summary, we show that three-platform sequencing offers many advantages over the existing NGS-based clinical sequencing workflows. Incorporation of WGS increased the yield of P/LP variants by 20% compared with combined WES and RNA-Seq, the current gold standard for precision oncology[52], improved the utility of RNA-Seq and provided highly accurate variant cross-validation. Higher WGS coverage would further improve sensitivity for detection of subclonal structural variations, which is important for samples with low tumor purity or high intra-tumor heterogeneity. In recognition of the value of experimentally-verified, multi-platform NGS data for developing future research and clinical applications, the raw and processed sequence data used in this study can be accessed via the St. Jude Pecan (Pediatric cancer) Data Portal in St. Jude Cloud at https://pecan.stjude.cloud/permalink/cts.

## Methods

**Patient samples selection**. We selected 78 pediatric cancer cases, treated at St. Jude Children's Research Hospital for a wide range of cancer types (Fig. 1b, Supplementary Fig. 1) with informed consent through our institutional review board (IRB) approved protocol in accordance with the Declaration of Helsinki (St Jude IRB# XPD09-018; PCGP). Paired tumor and normal samples were obtained either as fresh frozen tissue, genomic DNA or total RNA samples. Nucleic acids from all fresh frozen tissue samples were extracted (genomic DNA from both tumor and normal, total RNA from the tumor) and sequencing was performed according to standard operating procedures in our Molecular Pathology CAP/CLIA laboratory.

**Nucleic acid extraction and purification**. Before DNA or RNA extraction, leukemia samples with < 70% blasts or no available matched normal/remission tissue and sufficient material were flow-sorted based on antibody markers (CD45, CD3, CD34, CD19, and CD33) as determined by our clinical immunopathology tests. DNA was directly extracted from fresh frozen or flow-sorted tumor tissue using phenol-chloroform and treated with RNase A to remove residual RNA. RNA was extracted using TRIzol reagent (Thermo Fisher Scientific) and further purified using the RNAeasy Mini Kit (QIAGEN). DNA and RNA integrity was assessed by agarose gel electrophoresis and Agilent 2100 BioAnalyzer (Agilent Technologies), respectively. DNA and RNA concentrations were measured using a Qubit Fluorometer (Thermo Fisher Scientific).

**WGS, WES, and RNA-Seq library preparation and sequencing**. WGS libraries were constructed using the TruSeq DNA PCR-Free sample preparation kit (Illumina, Inc., CA) following the manufacturer's instructions. WES libraries were prepared using the TruSeq exome enrichment kit v1 (Illumina) per manufacturer's instructions, with some modifications (Supplementary Methods). For RNA-Seq libraries, we used a TruSeq total RNA protocol, as, unlike the TruSeq mRNA protocol, it can provide strand information using less RNA, even in sub-optimal samples. Total RNA-Seq analysis also enabled detection of both coding and non-coding RNA, along with other long intergenic noncoding RNA (lincRNA), small nuclear RNA (snRNA), and small nucleolar RNA (snoRNA). RNA-Seq libraries were constructed using the TruSeq Stranded Total RNA Kit, with Ribozero Gold (Illumina, Inc., CA) per manufacturer's recommendations, with slight modifications (Supplementary Methods).

After library quality and quantity assessment, each WGS tumor or normal sample was sequenced in three lanes of a v3 flow cell on a HiSeq2000 or HiSeq2500 instrument (Illumina, Inc, CA); WES samples were sequenced in pools of five to seven across three lanes and RNA-Seq samples were sequenced in pools of five to seven in one lane of a v3 flow cell (HiSeq2000/2500). Where necessary, additional sequencing (top off) was performed in Rapid mode sequencing (HiSeq2500) to ensure data analysis was completed in a clinically appropriate time frame.

**Hybrid Capture Validation**. Validation experiments were essentially as described previously[53]. Briefly, capture oligos were designed to encompass putative SNVs, SVs, and indels and used as input for NimbleGen Seqcap EZ solution bait sets. The library construction and target enrichment were performed per the manufacturer's instructions (Roche) using repli-G (Qiagen) WGA DNA. Enriched targets were sequenced on the same platform as above using paired-end 100-cycle sequencing. Putative variants were recovered using the same bioinformatics pipelines as below.

For Capture Validation (CapVal), we used the following algorithm to determine the validation status: First we obtained the mutant and reference allele counts for each marker in the corresponding sample, requiring a minimum base quality of 15 for SNVs and 5 for indels. A marker with 0 mutant reads and > 20 total reads in tumor sample is called "Wildtype". A Fisher's Exact test was then performed on the tumor and normal CapVal read counts. When mutant reads are present in the normal samples, markers with Fisher's Exact $P$-value $>= 0.05$ or $P$-value $<= 0.05$ and normal MAF $>= 0.2$ were called germline. Markers with $P$-value $<= 0.05$ are called "Somatic". To account for tumor-in-normal contamination (e.g. case SJHGG003_A), we required tumor MAF $> 0.05$ if mutant reads were observed in the normal sample for "Somatic" calls. In addition, we manually inspected markers with $P$-value greater than 0.05 and adjusted the validation status to "SOMATIC" for 16 SNVs and 4 indels (annotated with an asterisk (*) in Supplementary Data 6A and B, due to reduced coverage in germline DNA or tumor-in-normal contamination (e.g. IFFO1.M364fs in SJHGG003_A). Since homopolymer artifacts are frequently observed in WES/CapVal, we also required $>= 10$ mutant reads in the CapVal data if the homopolymer marker was discovered by WES only, regardless of the CapVal depth.

As a result, 88 indels were called "SOMATIC" after validation, of which 3 were from PCGP. We identified 89 indels from our Clinical pipeline, of which 84 indels were covered in CapVal, and 83 validated as "SOMATIC". This corresponds to a sensitivity of 94.3% (83/88) and a PPV of 98.8% (83/84). Similarly, 695 SNVs were called "SOMATIC" after validation, of which 34 were from PCGP. We identified 794 SNVs from our Clinical pipeline which included the cross-validation filtering process, of which 662 were covered in CapVal, and 653 were validated somatic. This corresponds to a sensitivity of 94% (653/695) and a PPV of 98.6% (653/662). For non-exonic SNVs (other SNVs), we are able to calculate PPV but not sensitivity because capture validation was performed only for variants detected and passed the filters in the clinical pilot study.

**Assessment of analytical performance**. Sensitivity and PPV were calculated using the set of variants that underwent capture validation and were covered after sequencing and alignment. We assessed PPV (true predicted positives divided by total predicted positives) rather than specificity (true predicted negatives divided by actual negatives) because the extremely high number of assessable positions inflates specificity beyond utility. For calculation of sensitivity, a variant was counted as a false negative in the clinical pilot study if it was not detected or was detected but filtered out by cross-validation.

In Supplementary Data 7A, variants that had support in a platform that was insufficient for detection but sufficient to cross-validate a call from another platform were labeled as Rescue. These variants are considered false negatives in the rescue platform alone.

**Data quality control**. Two main quality controls were used in the analysis pipeline. (i) Coverage QC used the percent of exons from a de-duplicated BAM file with average coverage greater than a sequencing-type-specific cutoff (see Results section). Samples initially failing QC had top-up sequencing performed. We did not impose a specific QC cutoff for duplicate reads, rather we based our coverage calculation on reads not marked as duplicates for all three platforms. (ii) The CONSERTING copy number profile plot was visually inspected for a highly fragmented copy number profile that we have seen as an infrequent but recurrent library preparation artifact.

**Mapping and SNV/indel calling**. All computational analysis happened on a dedicated compute and storage infrastructure designed and implemented by the High Performance Computing Facility at St. Jude. The analysis pipeline is outlined in Supplementary Fig. 9. We initially evaluated the sensitivity and PPV of our pipeline using the well-characterized COLO-829 cell lines[54] (Supplementary Methods). COLO 829 (ATCC® CRL-1974™) and COLO 829BL (ATCC® CRL-1980™), were obtained from ATCC (American Type Culture Collection, VA, USA).

DNA reads were mapped using the backtrack algorithm ("aln" and "sampe" steps) of BWA 0.5.9[55]. At the time the mapping was performed, BWA 0.5 was the latest stable version of BWA, as 0.6 and 0.7 were being released with frequent bug fixes. The changelogs of BWA versions 0.6 and 0.7 do not list any significant change to BWA backtrack. Aligned files were merged, sorted and de-duplicated using Picard tools 1.65 (broadinstitute.github.io/picard/).

RNA reads were mapped using our StrongARM pipeline, described previously[26]. Paired-end reads from RNA-seq were aligned to the following four database files using BWA: (i) the human GRCh37-lite reference sequence, (ii) RefSeq, (iii) a sequence file representing all possible combinations of non-sequential pairs in RefSeq exons and, (iv) the AceView database flat file downloaded from UCSC representing transcripts constructed from human ESTs.

Additionally, they were mapped to the human GRCh37-lite reference sequence using STAR. The mapping results from databases (ii)–(iv) were aligned to human reference genome coordinates. The final BAM file was constructed by selecting the best of the five alignments.

Gene-level read count was generated using HTseq-count[56] and FPKM value was calculated based on the transcript models in GENCODE v19.

We called SNVs and Indels in WGS and WES using Bambino[57], initially requiring that all reads were primary alignments and had a mapping quality of at least 1. The variant allele required a minimum of three supporting reads with a read quality 20 or better at the target site and within a ± 5 nucleotide window. If there were at least two reads supporting the variant, at least one of them must have had 10+ nt of flanking sequence.

**SNV/indel filtering and cross-validation.** As described previously, preliminary SNV/indel calls were filtered to remove those whose contributing reads aligned perfectly to another locus. Predicted indels were realigned using a Smith–Waterman approach[21]. The filtered variants were classified into high or low quality based on evidence of NGS read coverage and quality for supporting its presence in tumor and absence in the matching germline.

A variant was considered cross-validated if it met any of the following criteria: (1) automated detection by both WGS and WES; (2) automated detection by one platform (i.e. WGS or WES); the mutant allele had at least 1 supporting read in tumor and was absent in normal in the other platform; (3) automated detection by one platform as a high-quality variant; the mutant allele was absent in normal (read depth ≥ 20X) in the other platform; (4) automated detection in one platform as a high-quality variant with ≥ 20X coverage in normal; the mutant allele has at least 1 supporting read in RNA-seq. To extract read count from a sequencing platform where the mutation was not detected automatically, we implemented a "matrix" code which counts reads not flagged as optical/PCR duplicates and have base quality ≥ 15 at the queried variant site. These read counts were used for cross-validation.

Manual review was performed for exonic variants that passed the above-mentioned computation filtering. Non-exonic variants were not manually reviewed. Given the 2% difference on the validation rate between exonic SNVs (99%) which involves manual review and non-exonic SNVs (97%) which relies on the automated computational analysis, we estimate that manual review improves PPV by 2%.

All filtered variants were annotated using in-house scripts. Annotations were extensive and included an effect on protein coding sequence, population frequency, frequency in PCGP and the Catalogue Of Somatic Mutations In Cancer (COSMIC; http://cancer.sanger.ac.uk/cosmic) and various functional predictions. Hotspot mutations within BRAF at low depth were detected using clinsek tpileup (version 0.1; https://bitbucket.org/wanding/clinsek)[58].

**SV/CNV/fusion calling.** SV detection used CREST[23] and CNV detection used CONSERTING[24]. To improve sensitivity, CREST was run using four configurations: (i) standard tumor/normal, (2) tumor-only, (3) highly sensitive tumor/normal mode restricted to a configurable list of 43 genes known to be involved in pathologic fusions, (4) highly sensitive tumor/normal mode restricted to loci in the proximity of a copy number change point (CONSERTING-CREST)[24]. SV junctions were annotated according to read strand orientation and genes at breakpoints. Calling of CNVs and LOH was performed using CONSERTING under standard parameters and these results were used for purity estimation (see below). CNVs and SVs were used to assess chromothripsis following the guidelines by Korbel and Campbell[59].

CNV and LOH detection from WES used the Sequenza algorithm under recommended parameters[25]. This analysis was carried out as a research analysis. To estimate false negatives for sub-arm P/LP CNAs in WES (excluding iAMP 21 B-ALL samples), we took WGS copy number segments from CONSERTING in bed format and identified the segment overlapping or within the gene of interest. In the case of GLI2 amplification in SJMB030020, the gene was split over several distinct amplified copy number segments so we used the gene co-ordinates themselves as our region of interest. We next overlapped the Sequenza WES copy number segments bed file with our regions of interest and looked for any segment overlap requiring a log2-fold shift of ± 0.2 for gains and losses respectively in WES. However, even with a less stringent threshold of ± 0.1, no additional segment overlaps were identified. Potential segment overlaps between WGS and WES were manually reviewed to confirm that the effect on gene function e.g. one copy deletion, two copy deletion or amplification was consistent between WES and WGS. To identify potential false calls in WES, we identified WES segments with a log2 shift of ± 0.2 with no corresponding segment in WGS. We then annotated these segments for gene and cancer gene content using our clinical pipeline. We manually identified copy number segments worthy of committee discussion and potential classification.

Fusion detection from RNA sequencing used our in house Cicero tool (Y. Li et al. in preparation; source code is available upon request)—essentially the RNA-Seq equivalent to CREST[23]. ITD rearrangements were detected using Cicero in ITD mode on aligned RNA-Seq and WES data for loci known to undergo ITD (FGFR1, FLT3, PDGFRA, NOTCH1, KMT2A, EGFR, and PIK3R1). We also performed a FLT3 exon 14 hotspot check using Cicero-ITD on the FLT3 exon 14 regional reads from WGS.

We also developed a method to match the predicted DNA fusions with expressed RNA fusion products. Because of RNA splicing, fusion gene products typically carry different breakpoints between DNA and RNA. Briefly, a 5′-end breakpoint of a RNA fusion gene was linked to an SV in DNA if (1) the SV orientation supported the fusion product and (2) the DNA breakpoint was the same or downstream of the RNA breakpoint, up to the transcription end site. Similarly, a 3′-end fusion breakpoint in RNA was linked to a DNA SV when (1) the SV orientation supported the fusion product and (2) the DNA breakpoint was the same or upstream of the RNA breakpoint up to the transcription start site. The logic of linking method logic was validated using the oncogenic C11orf95-RELA fusion gene in ependymoma samples. While an existing program[60] failed to link the C11orf-RELA fusions from RNA and DNA samples, our method successfully linked these fusions in all fusion positive samples tested.

**Tumor purity estimation.** A loss of heterozygosity (LOH) score was calculated as the absolute difference between the allele fraction in tumor and germline sample for heterozygous germline SNVs. For example, an SNV from a region with a clonal single copy number loss and 100% tumor purity would have an LOH score of 0.5. We used LOH scores from SNVs within tumor copy-neutral LOH or heterozygous copy number loss regions (CNV value of between 0 and −1) to estimate tumor purity for all whole genome sequencing (WGS) samples. We assumed that the real genomic copy number of a single copy loss in $x$% of tumor cells could be estimated as $-x/100$ with an LOH value of $x/(400-2x)$. Assuming the remaining LOH signal came from CN-LOH (CN-LOH in $x$% tumor cell resulted in a LOH value of $x/(100 \times (2\text{-CNA}))$), the tumor content in a region could be estimated as the sum of the fraction with copy number loss and fraction with CN-LOH by: $2 \times ai + (CNA)/2 \times (1-2 \times ai)$.

Using tumor content estimates from various regions within the genome, we performed an unsupervised clustering analysis using the mclust package (version 3.4.8; https://cran.r-project.org/web/packages/mclust/index.html) in R (version 2.11.1; https://www.r-project.org/). The CNV/LOH based tumor purity of the sample was defined as the highest cluster center value among all clusters. We further ran the mclust algorithm on mutant allele fractions for somatic SNVs in diploid regions without LOH. The MAF based tumor purity of the sample was defined as the highest cluster center value among all clusters. The final tumor purity is defined as the larger of the CNV/LOH based estimate and the MAF-based estimate (X Chen et al. in preparation; code available upon request).

**Automated variant classification.** Two main data sources were included in auto-classification pipeline: COSMIC, and the St Jude GeDI database which hosts all somatic lesions including SNV/indels, CNAs, and SVs identified by the St. Jude/Washington University Pediatric Cancer Genome Project (available at https://pecan.stjude.org/#/home).

We first generated lists of genes recurrently mutated in cancer. To identify these genes, we focused on validated non-silent coding somatic mutations that had either an over-representation of truncation mutations (tumor suppressors) or a recurrent hotspot mutations (oncogenes). For COSMIC, we included only mutations which were experimentally verified and from genome-wide screens. To remove false positive sites that were germline polymorphisms, we further filtered those that overlapped with germline variants present in > 10 of the 6500 non-cancer individuals sequenced by NHLBI GO Exome Sequencing Project (ESP, http://evs.gs.washington.edu/EVS/). To mitigate the impact of hypermutable samples on global mutation profile, we also identified and removed hypermutated tumors (> = 100 coding variants and top 10% highest mutated tumors of their published study). PCGP samples were analyzed in a similar way and the resulting list was checked against the literature.

Genes affected by structural variations were ascertained from three different sources: (a) 23 fusion genes tested at the Molecular Diagnostic lab at St. Jude Children's Research Hospital; (b) Recurrent fusions identified by PCGP; and (c) 328 fusion genes resulting from chromosomal translocation identified by Cancer Gene Census (http://cancer.sanger.ac.uk/cancergenome/projects/census/). Genes affected by copy number abnormalities comprised of recurrent CNAs identified from PCGP, and genes with known amplification ($n = 16$) or deletion ($n = 38$) identified by Cancer Gene Census.

SNVs/indels affecting one or more of our 565 genes and all CNAs/SVs were flagged for manual review by our auto-classification software. Further, mutation types consistent with a gene's propensity to lose or gain function in cancer—for example, truncation of a tumor suppressor, hotspot mutation in an oncogene or amplification of a known gene target—resulted in an elevated review priority.

**Pathogenicity assignment.** We classified germline variants according to the previous recommendations[61]. However, as no consensus guidelines currently exist for the joint classification of somatic coding/structural/copy number changes, we classified these using a modified version of the same general scheme. Pathogenic (P) variants comprised of. (i) Hotspot SNV/indel mutations from any cancer type from known cancer genes (Supplementary Data 9). (ii) Nonsense/ frameshift/ splicing mutations in tumor suppressor genes where the gene was known to play a role in that cancer type. (iii) Recurrent gene fusions, deletions, truncations,

amplifications or arm-level abnormalities currently used for molecular diagnosis of that cancer type. Likely Pathogenic variants were: (i) Any mutation type in a gene linked to that cancer type but the functional impact of the change was unclear. (ii) Mutations with an obvious functional effect on a cancer gene but in a tumor type with no known association to that gene. Uncertain variants were those with an unclear functional impact in a cancer gene not known to play a role in that cancer type. However, rather than rigid adherence to these rules, our expert committee reserved the right to apply clinical, biological and functional insight from laboratory studies to support or overrule any given classification.

**Code availability**. We make extensive use of previously published algorithms as described above. Additional code used in this study is available upon request.

## Data availability

Raw and processed sequence data used in this study has been deposited in St. Jude Cloud and can be accessed via the St. Jude PeCan (Pediatric Cancer) Data Portal at https://pecan.stjude.org/permalink/cts. Raw data is available under accession SJC-DS-11003. Raw data is also available in EGA under accession EGAS00001002217.

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

## Acknowledgements

We thank Drs Cyrine Haidar, Kim Nichols, Armita Bahrami, Brent Orr, John Choi, Alberto Pappo, Mary Relling, Guo Zhu and Justin Baker for participating in expert review. We thank Drs. Susanne Baker and Charles Mullighan for providing DNA and RNA samples for a subset of the cases. We thank Drs. Kevin Shannon, Adolfo Ferrando and Charles Sherr for assisting interpretation of NRAS mutations, NOTCH1 ITD and CCND2 amplification. We thank Kim Nichols for critical review of the manuscript. We thank Paul Mead for designing and performing flow sorting. This work was funded by the American Lebanese Syrian Associated Charities of St. Jude Children's Research Hospital and National Cancer Institute grant P30 CA021765 (St. Jude Cancer Center Support Grant).

## Author contributions

J.R.D. conceived the study. J.Z., J.R.D., and D.W.E. designed and supervised the study with assistance from S.S., M.F.W. and T.A.G. M.R. designed and supervised computational pipeline development. J.N. and S.S. designed sequencing platform. J.N. and J.G. performed nucleic acid extraction, quality control, library preparation and sequencing. S.S., S.R., T.A.G., and R.G.T. assisted with case selection, provided molecular and cyto-genetic analysis and managed patient samples. M.R., A.T., and J.M. developed the computing pipelines/infrastructure. M.P., J.B., A.P., and E.H. developed the web interfaces and databases. J.Z., M.R., M.N.E. and Z.Z. wrote code for variant calling, annotation and classification. X.C. wrote code for purity estimation, linking DNA and RNA SV, CNA identification and classification. Y.Li wrote code for fusion gene prediction. X.M., Y.Li, X.C., and M.N.E. wrote code for variant validation. Y.Liu and S.N. performed exome CNA analysis. K.H. performed indel analysis. J.Z., Z.Z., M.P., X.M., D.H. and S.N. manually reviewed cases. S.N., Z.Z., D.H., M.P., Y.J., X.M., J.R.D., J.Z., J.N., S.S., M.F.W., S.R., R.M., G.W.R., and J.M.K. performed clinical interpretation. J.E., D.Y., and B.V. performed validation capture experiments. J.Z., S.N., and MR wrote the manuscript.

## Additional information

**Competing interests:** The authors declare no competing interests.

