## [Peer Review File · Nature Communications]

Reviewers' Comments:

Reviewer #1:

Remarks to the Author:

The manuscript by Rusch and colleagues addresses an important question on the benefits of including WGS to the existing CLIA NGS testing. The data is well presented though short in details in several areas. While the authors have taken a stance that including WGS is beneficial, the data presented does not seem to render convincing evidence along those lines. They should also provide a solid counter argument without any bias, unlike the small text presented in the discussion, on where and why one could skip WGS. Overall lack of analysis and lack of solid evidence to support the claims is discouraging and diminishes the importance of this study.

Major Issues:

1. Currently most ongoing precision oncology efforts do not incorporate whole genome sequencing or even whole exome sequencing due not only to time constraint, as noted by the authors, but also to cost-benefit considerations. The time factor is only superficially noted by the authors. They state in the Discussion "Presently, generation of a curated mutation report from raw Total Sequencing data takes approximately 8 days, making it possible to achieve our ultimate goal of returning results in between 28 and 42 days from initial sample acquisition." The authors should indicate the actual number of days it took them "from biopsy to the final report" and compare against equivalent timeframe reported in other clinical sequencing efforts to provide a realistic sense of the extra time involved.

2. Cost is a consideration as well, including not only sequencing cost but also data-storage and analysis costs. Even as a systematic analysis may not be in the scope of the present study, some indication of cost with and without WGS/WES rather than targeted gene panel should be provided for reference.

3. A key claim of the study is summarized in the discussion as "We showed that, independently, WGS detected 90% of P/LP variants compared to only 35% by WES and 20% by RNA-Seq". This huge discrepancy is possibly accounted for in a large measure because the authors do not use WES data for CNA analyses. The reason cited for this is not convincing at all and almost certainly not shared by the NGS community at large. Numerous published clinical sequencing studies have routinely used WES, and much more commonly, targeted gene panel sequencing, to assess and report clinically actionable CNAs (Ortiz, *Pediatr Blood Cancer* 2016; Hyman, *Drug Discovery Today* 2015; Cheng, *J Mol Diag* 2015; Robinson, *Cell* 2015; Mody, *JAMA* 2015; Hovelson, *Neoplasia* 2015; Roychowdhury, *Sci Trans Med* 2011; Frampton, *Nat Biotechnol* 2013; Ross, *J Clin Pathol* 2014; Ross, *JAMA Oncol* 2015; He, *Blood* 2016; Lee, *J Clin Pathol* 2016; Ross, *Cancer* 2016; Wheler, *Cancer Res* 2016; Chmielecki, *Cancer Res* 2017). CNAs are eminently detectable by WES/ targeted panels, except for arm level/whole chromosome copy number aberrations, which are also amenable to analyses with larger targeted panels (Lonigro, *Neoplasia* 2011; Pritchard, *J Mol Diagn* 2014). For a fuller assessment, the authors should perform CNA analysis with WES data and highlight the clinically actionable/ informative CNAs that could be called only by WGS.

4. The depth of coverage for WES in this study is pegged at >100x on average for both normal and tumor samples (Methods, Results, Supp Table 3). This is at the low end of coverage for clinical sequencing- particularly for samples with low tumor content. Interestingly, in Table 3, 29 tumor samples are noted with less than 100X coverage for WES. Of these, 5 samples have tumor content less than 50%, not specified for another 4 samples. Since clinical sequencing by definition pertains to analysis of individual cases, limited coverage of so many cases cannot be reconciled by averaging over the entire cohort. Also, this limited coverage rather than any intrinsic limitation of WES data over WGS data may account for some of the noted lack of sensitivity. Increasing the WES coverage is relatively less demanding than WGS to compensate for it.

5. Detection of internal tandem duplications (ITD) such as in FLT3 or NOTCH are highlighted as

problematic and analyzed using an ITD specific Cicero data analysis tool. However, FLT3-ITD, as well as other informative indels, have been reported through various standard somatic indel analysis pipelines using targeted exome panels (Mody, JAMA 2015; He, Blood 2016; Chmielecki, Cancer Res 2017). Furthermore, the detection of indels can be corroborated with parallel analyses of RNA sequencing data and does not need WGS.

6. WGS data is cited as a useful companion for detection of gene fusion break-points. As an example, PDGFRA gene fusion in a high grade glioma is cited (Figure 2b). However, after a description of complex rearrangements involving three chromosomes, it is mentioned that in this case PDGFRA is amplified with ~22 copies. In this context, the significance of the gene fusions is not clear. Can the authors comment as to what fraction of PDGFRA expression is accounted for by the fusion? Or possibly, these fusions merely represent by-products of focal amplification of PDGFRA and are thus not particularly relevant. The authors should point out if this sample displays outlier expression of full length PDGFRA or only the chimeric transcript.

7. In the Table 6 summarizing “all pathologic findings”, the authors should include coverage/read support for SNVs and indels detected in PGCP/current study to help assess if sensitivity/specificity of detection is simply contingent upon coverage, not an intrinsic limitation of the sequencing platform.

8. Among “novel findings due to the inclusion of WGS”, the GLI2 amplification, TP53 indel or the DNMT3A deletion (Figure 6), or Notch 1-ITD (Supp Figure 7) do not strike one as inherently undetectable by WES or even targeted panels. On the other hand, the failure to detect high level amplification of MYCN in a retinoblastoma sample is quite intriguing. The fact that the clinical sample analyzed here was different from the PGCP sample could be a plausible explanation, but is extenuated by the “almost perfect agreement” in the CNAs observed on chromosome 13 (Figure 5b).

9. The authors need to perform copy number analysis on WES data to make a fair comparison. They can perhaps try either “Falconx”, PureCN (PMC5157099) or Sequenza”, and see what works best. Though they cite an old reference (ref 37) in the discussion as a reason for not performing this analysis, several independent groups have been able to obtain this information from WES. Hence without these numbers from WES, the key data presented in Figure 5a (direct comparison between platforms) is highly misleading.

10. The failure to detect FXR1-BRAF in a low grade glioma in the WGS data due to low tumor content is understandable, however, it is not mentioned if this fusion was not detected in RNA-seq data as well. That would be a bigger concern. Elsewhere, it is concerning that fusions including KMT2A-MLL2, ETV6-RUNX1, KIAA1549-BRAF etc. were not detected. Can the authors provide data to support the claim that this is due to low expression and not suboptimal yield of RNA-seq data?

11. According to Supplemental Table S6 only two SNV/indels were not detected by WES (SH2B3 and FLT3). So in terms of the SNVs/Indels detection, WES performance is comparable to WGS in terms of LP/P events detection. The big difference then arises only because of amplification and deletions and SVs called from WGS, for which as stated above the numbers are not provided in this manuscript as CNV analysis are not done for WES.

12. Sample SJMLL001 WES data has very high PCR duplication rate, probably due to sample quality issue, though the WGS library has good WC, and perhaps is the reason why FLT3 mutation was not detected by WES. Did the authors try repeat the WES library prep? What is the reason for low coverage in the SH2B3 indel sample (SJBALL021900)? Can the authors show the coverage in these regions?

13. The WES indel/SNV callers seem to give consistently higher number of calls than their WGS

which is surprising (Suppl Figures 3 and 4). Is there a particular reason for this? Was the WES caller less stringent? This systemic discrepancy is probably reflected in the table in Suppl Figure 4b.

14. It appears that most notable SVs presented in the figures are all supported by fusion evidence from RNAseq. Here again while WGS does provide the evidence/details of the underlying genomic event, whether it adds to the calls among actionable/P/LP over RNAseq is not clearly presented?

15. Why did the authors use older versions for several programs? For example, BWA current version is 7.2 and Picard current version is 2.92. Did the authors see any difference between these versions?

16. From a per patient perspective, how many of the 78 cases benefit from WGS? Can the authors discuss this in detail? This could very well be the basis of recommending a multi-tiered CLIA/sequencing approach where cost/time can be reduced for reporting. For example, WES (Tumor/normal) and RNAseq (Tumor) can be performed in the first round. Samples with more complex genome can be identified and recommended for WGS.

Reviewer #2:

Remarks to the Author:

Rusch et al provide their experience of genomic profiling by total sequencing of whole-genome, whole exome, and whole transcriptome. The group has pioneered adoption of the advanced integrative next-generation sequencing to the field and there is inherent value in publishing their experience as more groups move along this path. There are several concerns of presentation that need to be addressed. The paper at times the paper reads as a branding or marketing presentation for a branded "Total Sequencing" which is unfortunate given the insight the results could provide and given total sequencing is a loaded phrase. The reporting of statistics was unclear and unfortunately seemed inconsistent. There was also insufficient discussion of the impact of sequencing depth on sensitivity, precision, and false negative rate given the low coverage sequencing typically expected for lower tumor content samples.

Major Concerns

Area 1. I found the branding of "Total Sequencing" to be unnecessary, ambiguous, and misleading given the phrase is highly loaded. It makes the paper ambiguous – should I review it on whether this is "total sequencing" or should I review the method "Total Sequencing"? I am choosing not to review on whether this is total sequencing, since it so clearly is not total sequencing, but rather optimized sequencing for deep exonic regions, mRNA, and relatively low pass cancer WGS. If the authors insist on it, could they more clearly (such as by a trademark symbol) distinguish this as branding? Even within regions that are well covered, triplicate expansions, inversions, and unphased DNA are just a couple of examples of categories that are missed. For example, there are several well-known repeat expansions that would be expected in these samples – the methods used do not manage expansions. The paper ends with Higher WGS coverage would further improve – which undermines total sequencing.

Area 2. I found the summary statistics very confusing and inconsistent, and at times it seemed as selective reporting. Typically, sensitivity, specificity, and precision are provided in one table. These values are presented in a series of figures and tables. However, it seems the reportable range is changing. It is best to present these such that one can assess these values together, such as to avoid biases through selection of reportable range idea for each measure. For example, 4A appears to have 84 indels. Figure 3 (Precision) has 112 e-indel. I'm sure it's consistent, but this is some other examples have me wondering what the values are.

There needs to be more detail on the calculation of sensitivity, specificity, precision, FDR. Are the metrics cited in the abstract presented for other variant classes/types. For example, synonymous variants, intronic variants, transcription start-site variants, and many other classes that are critical part of discovery and these do not seem to be considered in the reported accuracy measurements. How were true negatives determined? For example, in the cases where the tumor content was below 10% one would expect a large number of false negatives. Very specifically, the reportable range and the means by which true negatives were estimated need to be described. I wasn't sure for example if a manual rescue represents a true negative, false negative, or no result.

Area 3. Coverage is a critical aspect that is not quantitatively assessed in terms of a classic limits of detection analysis. The authors provide specificity and sensitivity, and discuss in great detail the importance of coverage. In practice for many clinical sequencing in CAP/CLIA environments the coverage they are sequencing is quite low and prone to false negatives. The various excel files highlight numerous examples where estimated tumor purity is less than 40%, less than 20%, and less than 10. The other concern is that the paper puts a major focus on whole-genome sequencing at relatively low coverage (30x) for mixed tumor/normal specimens, without going into the loss of sensitivity. Most clinical labs focusing on tumor sequencing often sequence to 400x -1,000x depth to examine lower tumor purity. The reported samples have a large portion of lower tumor content (50% are less than 50% tumor content), and so this discussion is relevant here. The authors should describe the dependence of sensitivity on tumor content of example.

Reviewer #3:

Remarks to the Author:

The authors performed "total sequencing", which includes whole genome (WGS), whole exome (WES) and whole transcriptome (WTS) sequencing, of 78 pediatric cancer patients. Their analysis pipeline detects, integrates, and cross-validates variants from each sequencing technique (platform). Experimental validation was performed to measure the accuracy of "total sequencing". It claims that there is a critical need for using WGS in cancer screening in conjunction with WES and WTS.

The cross-validation method itself is not entirely novel, e.g. using WES to validate WGS (<https://www.ncbi.nlm.nih.gov/pubmed/22178993>) or using SV predictions to validate RNA fusions (<http://www.sciencedirect.com/science/article/pii/S0092867412010227>). The article maybe of interest to the clinical community to learn more about the utility of sequencing using multiple techniques. In the research community it is not an uncommon approach as cost is not a major concern but discovery unlike in a clinical setting that diagnostics have to be done routinely and have to be interpretable/actionable.

It is commonly known to the community that the biggest limitation of WES and gene panel is on SV/CNV detection. There were also numerous suggestions to complement WES with WGS to fill that gap; e.g. using a low-cost, shallow WES for SV-driven tumors ([http://www.cell.com/trends/genetics/abstract/S0168-9525\(16\)30070-1](http://www.cell.com/trends/genetics/abstract/S0168-9525(16)30070-1)). On the other hand, WES or gene panel provides a more economic solution to sequence to very high depth (thousands of coverage) to detect low allele frequency variants in cancer, particularly in highly non-pure samples such as ctDNA. In that regard, Fig 4a in the article also showed that WES-only approach has higher sensitivity than WGS-only approach. Adding another sequencing can certainly validate the variants from WES; however, it is unclear in the article that how much more value it is for further validating detected, clinically actionable/interpretable variants which are usually having very high sensitivity and specificity in diagnostic assays. For other or rare variants that can only be detected by WGS, it is not clear how interpretable they are, as previously described e.g. in <https://www.ncbi.nlm.nih.gov/pmc/articles/PMC4484824/>.

Reviewer #1 (Remarks to the Author):

The manuscript by Rusch and colleagues addresses an important question on the benefits of including WGS to the existing CLIA NGS testing. The data is well presented though short in details in several areas. While the authors have taken a stance that including WGS is beneficial, the data presented does not seem to render convincing evidence along those lines.

[Response] We thank the reviewer for their comments. We have re-worked areas of the manuscript to highlight the value that WGS adds as part of a multi-platform test. Replies to specific points with examples are discussed below.

They should also provide a solid counter argument without any bias, unlike the small text presented in the discussion, on where and why one could skip WGS. Overall lack of analysis and lack of solid evidence to support the claims is discouraging and diminishes the importance of this study.

[Response] To address the reviewer's concern, we have performed a combined WES and RNA-Seq analysis to develop a solid counter argument against the three platform (WGS/WES/RNA-Seq) analysis described in the manuscript. This included the copy number variations obtained from WES, an analysis which was not performed in the prior version. We then summarized the performance of WES+RNA-Seq in the updated version of **Figure 5a**.

Major Issues:

1. Currently most ongoing precision oncology efforts do not incorporate whole genome sequencing or even whole exome sequencing due not only to time constraint, as noted by the authors, but also to cost-benefit considerations. The time factor is only superficially noted by the authors. They state in the Discussion "Presently, generation of a curated mutation report from raw Total Sequencing data takes approximately 8 days, making it possible to achieve our ultimate goal of returning results in between 28 and 42 days from initial sample acquisition." The authors should indicate the actual number of days it took them "from biopsy to the final report" and compare against equivalent timeframe reported in other clinical sequencing efforts to provide a realistic sense of the extra time involved.

[Response] We acknowledge that the analysis of turnaround time is limited in our current study. In a typical clinical genomics workflow, individual cases are analyzed end-to-end either individually or in small batches, without interruption. Analysis of our clinical pilot samples was performed during the development of the automation infrastructure. As such, the analysis was performed in large batches and was not performed end-to-end. Therefore, we were unable to directly measure turnaround time. However, we inferred turnaround time by adding the time spent on the individual analyses.

To provide statistics based on direct measurements of turnaround time, we gathered data

from our current clinical genomics service - which uses the same pipeline described in this study. We have added this information to the discussion. We are now confident that this inferred time is accurate, as we are presently able to turn around samples in a median of 31 days from biopsy/surgery to report. This compares fairly well to published reports listing tumor exome analysis in 2-5 weeks (Rennert et al. 2016), and < 21days for integrated panel/selected transcriptome (Zehir et al. 2017). We have incorporated the following text in the **Discussion** section, **p.20 paragraph 2**:

Generation of a curated mutation report from raw sequencing data was estimated to take 8 days based on batch analysis of the samples used in the study. Allowing for time library preparation and sequencing, we extrapolated a clinical turnaround time of 28-42 days. We have since deployed three-platform sequencing as a clinical service and achieved a median turnaround time of 31 days from sample receipt to report, which is comparable to a median turnaround time of 3 weeks reported in two recent clinical genomic studies that employed tumor exome sequencing (Rennert et al. 2016) or integrated panel/selected transcriptome sequencing (Zehir et al. 2017).

2. Cost is a consideration as well, including not only sequencing cost but also data-storage and analysis costs. Even as a systematic analysis may not be in the scope of the present study, some indication of cost with and without WGS/WES rather than targeted gene panel should be provided for reference.

[Response] Since actual costs were not measured during the study, we estimated cost based on data volume, reagent costs, and cloud storage costs in the revision. Analytical costs were not evaluated since all analysis was performed using local resources owned by the institution. As such, there were no direct charges for the analysis. It would be extremely difficult to accurately estimate actual cost based on amortizing capital expenditures and data center operating costs, so such an analysis was considered to be out-of-scope for this study. We have, however, incorporated the following text in the **Discussion** section, **p.21 paragraph 3**:

The disadvantages of three-platform sequencing are the computing infrastructure required for data storage and analysis, and the cost of sequencing. Data retained from each paired patient sample occupies 150-200 Gb and, in line with our clinical laboratory protocols, must be rapidly accessible for 6 months which is estimated to cost \$25 per case in a cloud computing platform. For this pilot study, sequencing was carried out using Illumina HiSeq 2000/2500 machines at a cost of \$25,000/case (\$18,000 for 30x WGS). However, costs have since been reduced to \$8,600/case (\$6,600 for 45x WGS) on our clinical service as samples are now sequenced by HiSeq 4000 machines. We anticipate that technology development will further reduce the cost for both computing and sequencing. As treatment of a pediatric cancer typically costs hundreds of thousands of dollars, the precision of integrative genomic profiling offered by three-platform sequencing outweighs the extended time for secondary validation and the potential for missing ~20% pathogenic variants in pediatric cancer.

We anticipate that technology development will further reduce the cost for both computing and

sequencing. For example, the estimated cost of three-platform sequencing using the most recent Illumina NovaSeq will be \$3,800-\$6,500 per case. This was not included in the text, as it was calculated from reagent costs provided by Illumina which are unpublished.

3. A key claim of the study is summarized in the discussion as “We showed that, independently, WGS detected 90% of P/LP variants compared to only 35% by WES and 20% by RNA-Seq”. This huge discrepancy is possibly accounted for in a large measure because the authors do not use WES data for CNA analyses. The reason cited for this is not convincing at all and almost certainly not shared by the NGS community at large. Numerous published clinical sequencing studies have routinely used WES, and much more commonly, targeted gene panel sequencing, to assess and report clinically actionable CNAs (Ortiz, *Pediatr Blood Cancer* 2016; Hyman, *Drug Discovery Today* 2015; Cheng, *J Mol Diag* 2015; Robinson, *Cell* 2015; Mody, *JAMA* 2015; Hovelson, *Neoplasia* 2015; Roychowdhury, *Sci Trans Med* 2011; Frampton, *Nat Biotechnol* 2013; Ross, *J Clin Pathol* 2014; Ross, *JAMA Oncol* 2015; He, *Blood* 2016; Lee, *J Clin Pathol* 2016; Ross, *Cancer* 2016; Wheler, *Cancer Res* 2016; Chmielecki, *Cancer Res* 2017). CNAs are eminently detectable by WES/targeted panels, except for arm level/whole chromosome copy number aberrations, which are also amenable to analyses with larger targeted panels (Lonigro, *Neoplasia* 2011; Pritchard, *J Mol Diagn* 2014). For a fuller assessment, the authors should perform CNA analysis with WES data and highlight the clinically actionable/informative CNAs that could be called only by WGS.

[Response] To address reviewer’s concern on lack of CNA analysis on WES, we used the Sequenza algorithm (Favero et al. 2015), as recommended by the reviewer in comment #9, to call copy number segments from WES; this is now described on **p.9 paragraph 2** and in the **Methods** section on **p.29 paragraph 2**. We compared P/LP copy number segments and LOH regions identified by WGS with those derived from WES and incorporated results into an updated version of Supplementary Table 6 (**Supplementary Table 7A** in the revised manuscript).

Our analysis showed that the WES could reliably identify arm-level abnormalities as well as many sub-arm changes, increasing the diagnostic yield of WES alone or WES+RNA-Seq to 60% and 78% of P/LP variants, respectively. We incorporated CNAs found by exome into a new version of **Fig 5** (shown below in our response to Reviewer 1’s comment #9 which concerns ITDs).

However, WES-based CNA analysis tended to miss some of the focal abnormalities. We compared CNAs that were missed by WES (i.e. false negative calls) with those that were detected by both WGS and WES in a new **Supplementary Figure 7**, shown below. CNAs missed by WES but found by WGS generally spanned smaller genomic regions and contained fewer exons. This suggests that CNAs affecting small and/or exon-poor regions would be challenging to detect using WES due to insufficient markers for robustly measuring read-depth changes - even if the algorithm utilizes off-target reads. WGS, on the other hand, is able to detect these CNAs as the algorithm we developed tallies 100-bp windows genome wide for measuring read-depth change (Chen et al. 2015).

Supplementary Figure 7. Comparison of P/LP sub-arm CNAs detected or missed by WES CNA analysis. Each panel shows the distribution of CNAs detected by both WGS and WES (Found; n=18) or detected by WGS and missed by WES (Missed, n=36) **a)** Distribution of CNA genome size (log2 of segment size in basepairs) **b)** Distribution of number of exons within CNAs **c)** Distribution of the number of basepairs of exonic regions covered by the CNA. Wilcoxon Rank Sum test showed significant differences in the distributions with p-values of p-value = 0.002253, 0.003404 and 0.005525 respectively.

We include changes to the diagnostic yields of WES and WES+RNA throughout the manuscript and specifically on **p.15 paragraph 3**, along with some notable misses with the following text:

We compared three platform sequencing with the current gold standard of combined WES and RNA-Seq. For the purposes of this study, we evaluated the approach on research basis rather than as a full CLIA validation. Combined WES and RNA-Seq identified 78% (188/242) of P/LP variants. The lower yield relative to three-platform sequencing was mostly driven by focal and exon-poor CNVs (n=36) that were not detected in our exome-only copy number analysis (**Supplementary Fig. 7**,

Supplementary Table 8). Notable misses included focal and intragenic disruptions of CDKN2A, CREBBP, ETV6, MYB, NF1, NR3C2, PAX5, PTEN, RB1 and TP53 (**Supplementary Table 7A** and **Supplementary Fig. 7**).

An additional challenge in WES-based CNA analysis is the lack of supporting evidence from structural variation breakpoints to corroborate read-depth changes; therefore, it is difficult to differentiate true positives from false predictions when a CNA has a low magnitude of read-depth change or has a small segment size. Of the twelve additional committee-reviewable CNAs identified by WES, none were verified by WGS. These likely false positive CNAs, along with information of detected and missed CNAs are recorded in a new **Supplementary Table 8**.

4. The depth of coverage for WES in this study is pegged at >100x on average for both normal and tumor samples (Methods, Results, Supp Table 3). This is at the low end of coverage for clinical sequencing- particularly for samples with low tumor content. Interestingly, in Table 3, 29 tumor samples are noted with less than 100X coverage for WES. Of these, 5 samples have tumor content less than 50%, not specified for another 4 samples. Since clinical sequencing by definition pertains to analysis of individual cases, limited coverage of so many cases cannot be reconciled by averaging over the entire cohort. Also, this limited coverage rather than any intrinsic limitation of WES data over WGS data may account for some of the noted lack of sensitivity. Increasing the WES coverage is relatively less demanding than WGS to compensate for it.

[Response] We thank the reviewer for suggesting an evaluation on false negatives on WES versus WGS coverage, which will be useful to determine whether low-coverage WES data or WGS platform was the culprit.

First, we would like to clarify that tumor purity for all 29 samples with <100X WES coverage were recorded on column M of the original **Supplementary Table 3**. As the same column was used to present both tumor purity of the tumor specimen and tumor-in-normal contamination of the germline specimen, it may have caused confusion regarding the purity status. We have modified this table so that tumor purity is now presented in column Q and the proportion of tumor-in-normal contamination in column I. The reviewer should be able to see that the tumor purity for the remaining 24 samples ranged from 0.62 to 1.00.

We focused the evaluation of WGS-only findings on point mutations (e.g. SNV and indels) as they were routinely analyzed by both platforms while performance of CNA analysis by WGS versus WES has already been presented in our response to the 3rd point raised by reviewer 1. Of the 7 P/LP variants missed by WES, none was present in the five samples (e.g. SJDSRCT030016, SJIFS030022, SJLGG026, SJLGG036, SJMPNST030013) that had low tumor purity and <100X WES coverage. Of the 7 variants missed by WES, 3 had low coverage at the site uniformly across all WES samples and we incorporated the cohort-level coverage statistics along with local GC content of these variants (e.g. SH2B3 R216fs, FLT3 D835Y, and CEBPA E59*) in a new **Supplementary Figure 3**, shown below. As the average of WES read count is approaching 0 across all samples, it shows that these variants were missed due to the capture efficiency of WES instead of coverage. Of the remaining four variants, NF1

T2263_Y2264fs was missed due to diminished capture efficiency for the mutant allele (only one read for mutant allele present in WES despite an average WES coverage of 106X compared to 14 in WGS with an average WGS coverage of 38X). Missing of NRAS Q61K, TET2 A1381T and ASXL1 R693* were caused by lack of sufficient high-quality reads in WES at the variant sites.

Supplementary Figure 3. WGS and WES coverage of three genes with P or LP SNVs or indels missed by WES. Plots show the WGS (blue) and WES (gray) coverage, averaged over all 78 germline samples, along the genes *FLT3*, *SH2B3*, and *CEBPA*. The gene model and GC content are depicted below the coverage graph, as in **Supplementary Figure 1**. Arrows indicate the location of P or LP SNVs/indels that were found by WGS but missed by WES. Each of these locations is well covered in WGS but systematically uncovered in WES.

We also performed a “reverse” experiment by evaluating 13 point mutations that were missed by WGS and found by WES. These WES-only variants were all cross-validated by WGS and only 3 out of 13 have mutant allele fraction (MAF) above 0.1 (range 0.01-0.17). Therefore, increasing WGS coverage may increase the power for detecting these variants in WGS. We have included a brief synopsis of this trend in the section of “Detection of diverse types of pathogenic and likely pathogenic variants” on **p.13, paragraph 1**. We expanded this discussion in **Supplementary Note 2**, entitled “Pathologic and likely pathologic mutations not detected by WES”:

A total of seven pathologic or likely pathologic somatic SNVs/indels were discovered by WGS alone. Of those seven two (*FLT3* and *SH2B3*) had no support in WES, and five had insufficient support in WES for de novo detection (**Supplementary Table 7C**). We manually inspected each of the seven to determine the cause for non-detection in WES. Four out of the seven sample also had whole exome sequencing from various research studies. For these four, we combined the reads from the clinical and research experiments and ran the resulting data through the analytical pipeline to determine if additional coverage would allow the variants to be detected

by WES. The results are given in **Supplementary Table 7C**. In brief, three of the seven were in regions of systematically low coverage and would therefore be unlikely to be recovered by additional coverage (one had additional reads, and it was not detected with additional coverage). One appeared to be caused by poor capture of the indel-harboring fragments and would also be unlikely to be recovered by additional sequencing. The other three showed no signs of systematic WES-related problems and we all recovered with additional reads.

5. Detection of internal tandem duplications (ITD) such as in FLT3 or NOTCH are highlighted as problematic and analyzed using an ITD specific Cicero data analysis tool. However, FLT3-ITD, as well as other informative indels, have been reported through various standard somatic indel analysis pipelines using targeted exome panels (Mody, JAMA 2015; He, Blood 2016; Chmielecki, Cancer Res 2017). Furthermore, the detection of indels can be corroborated with parallel analyses of RNA sequencing data and does not need WGS.

[Response] In the original manuscript, we stated that “ITDs ranged from 24 to 96 bp were detected by several different combinations of WGS (either as an indel or a CNA), WES and/or RNA-Seq”. This statement does not imply that ITD could only be detected by WGS or RNA-Seq. Indeed, CICERO on WES found 6/7 P/LP ITDs including 5/5 FLT3-ITDs. This result was not included in the original submission, but in the revised manuscript we have included WES CICERO results in column K of **Supplementary Table 7A**, and we also reported WES ITD results on **p.14 paragraph 2**. The diagnostic yields shown in a revised **Fig. 5** also reflects this addition and we have included a comment in the **Methods** section on **p.29, paragraph 2** stating that WES ITD analysis as part of the analytical process.

We do want to reiterate our observation that not all ITDs can be detected by WES or by standard indel detection methods. Specifically, WES may miss large ITDs that span multiple exons (e.g. the kinase domain ITD of FGFR1 in low grade glioma, Zhang et al. 2013) or ITDs that span an intron-exon boundary. The NOTCH ITD highlighted in the section of “Novel findings due to the inclusion of WGS” was such a case. In the revised manuscript, we clarified on **p.17, paragraph 3** that the mutant NOTCH1 ITD allele was entirely absent from the raw exome capture data, thus cannot be detected regardless of the computational analysis approach used (e.g. CICERO ITD or standard indel analysis such as Pindel).

We next explored the Reviewer’s suggestion of using WES and RNA-Seq cross validation as a strategy for detecting ITDs and indels. Comparing genome-wide Cicero-ITD predictions from WES and RNA-Seq identified 275 and 620 potential ITDs respectively. However, only a small minority of predictions (7) had cross platform support and these included all P/LP ITDs except for NOTCH1 (discussed above).

We explored the same WES+RNA-Seq cross validation strategy for indel detection. For the 31 P/LP/U indels detected by WGS and/or WES listed in **Supplementary Tables 7A/B**, only 25 could be cross-validated using RNA-Seq either through a directly equivalent mutation call (11/25) or a “fuzzy” match with a slightly different indel being called (6/25) or manual inspection and rescue of

aligned RNA-Seq reads (8/25). The remaining 6/31 indels (19%) had no supporting RNA evidence. From this analysis, we conclude that indel calling from exome with cross-validation from RNA-Seq is a viable strategy but may be more susceptible to false negative calls than three platform sequencing due to low expression/nonsense mediated decay of the target gene. We include variant allele counts from RNA-Seq for P/LP variants in **Supplementary Table 7B, column R and explanatory notes in column U.**

6. WGS data is cited as a useful companion for detection of gene fusion breakpoints. As an example, PDGFRA gene fusion in a high grade glioma is cited (Figure 2b). However, after a description of complex rearrangements involving three chromosomes, it is mentioned that in this case PDGFRA is amplified with ~22 copies. In this context, the significance of the gene fusions is not clear. Can the authors comment as to what fraction of PDGFRA expression is accounted for by the fusion? Or possibly, these fusions merely represent by-products of focal amplification of PDGFRA and are thus not particularly relevant. The authors should point out if this sample displays outlier expression of full length PDGFRA or only the chimeric transcript.

[Response] In the original manuscript, Fig. 2C showed that amplification only affected part of PDGFRA but did not state explicitly that the 3-chromosome SV was subsequently amplified resulting in a 22-66 copy fusion gene amplicon. In the revised manuscript **p.9, paragraph 3**, we clarified this point by stating that the amplicons only encompass the 3' end of PDGFRA with the 5' end being non-amplified by incorporating the following:

An example of integrative DNA-RNA SV analysis is demonstrated in the detection of a complex *PDGFRA* gene fusion in a high-grade glioma (**Fig. 2b**). This sample contained a high-level complex amplification that included exons 10-23 of PDGFRA with exons 1-9 being excluded from the amplified region.

The fusion lacked the PDGFRA extracellular domains encoded by exons 1-9. We also revised the display of PDGFRA protein domains in **Fig. 2b** based on a published data (Dai et al. 2013) to make the extracellular and juxtamembrane domains more evident. The revised **Fig. 2b** also labels the number of wild-type RNA splice junction reads at PDGFRA exon 9-10 (n=727) in parallel with the number of fusion junction reads linking DIP2C exon 1 to PDGFRA exon 10 (n=33), exon 11 (n=149), and exon 12 (n=26). This shows, surprisingly, that the expression of amplified region of PDGFRA is not strongly correlated to the level of DNA amplification (estimated to be 22-66 copies). This was because fusion gene expression was driven by the DIP2C promoter, and wild-type DIP2C expression is much lower than wild-type PDGFRA (approximately 10%, of PDGFRA; data not shown). As all of the fusion isoforms contained the exon 12-encoded PDGFRA juxtamembrane domain but lacked intact extracellular domains, this suggests that DIP2C-PDGFRA fusion is likely to function as a constitutively active kinase similar to

the known KDR-PDGFR Δ 8,9 previously reported in glioblastoma (Ozawa et al. 2010). We hypothesize that, because the fusion transcript was driven by the weaker DIP2C promoter, high-level of amplification of DIP2C-PDGFR Δ 8,9 ensured the fusion protein was present at a sufficient level to be oncogenic. We incorporated these points into the following **Supplementary Note 3** entitled “Expression of DIP2C-PDGFR Δ 8,9 fusion gene”:

As shown in **Figure 2b**, the PDGFR Δ 8,9 amplification encompassed only the 3’ end of PDGFR Δ 8,9 and omitted the extracellular domains encoded by exons 1-9. Counting of wildtype and fusion junction reads from RNA-Seq showed 727 wild-type exon 9-10 junctions, implying expression of wild-type PDGFR Δ 8,9 was higher than that of fusion gene whose RNA junction reads numbered 26 and 149 from DIP2C exon 1 and PDGFR Δ 8,9 exons 10 and 11 respectively. This showed, surprisingly, that the expression of amplified region of PDGFR Δ 8,9 was not strongly correlated with the level of DNA amplification. Further investigation showed that DIP2C has generally low expression in High Grade Glioma with FPKM of <14 in non-amplified PCGP samples (data is available at <https://pecan.stjude.org/proteinpaint/DIP2C>). Given that the weak DIP2C promoter drove fusion gene expression, we hypothesize that amplification was necessary to achieve a sufficient level of DIP2C-PDGFR Δ 8,9 for oncogenic action. Similar rearrangements of PDGFR Δ 8,9 including KDR-PDGFR Δ 8,9 and PDGFR Δ 8,9 that lack intact extracellular domains show PDGFR Δ 8,9 kinase activation (Ozawa *et al.* 2010) and appear to be oncogenic even at relatively low expression levels. For example, Brennan et al. (2013) used a >10% of total PDGFR Δ 8,9 expression to call a sample positive for the rearrangement in their analysis of RNA-Seq generated from glioblastoma samples from The Cancer Genome Atlas (TCGA) project.

We also include here an excerpt from **Fig. 2** showing the above-mentioned annotations:

Figure 2b. The fusion transcripts are shown in protein view (b) with the domains marked in color and the vertical dotted lines marking the boundaries of each exon with chimeric RNA read counts indicated above the PDGFR Δ 8,9 ideogram and wildtype exon 9-10 read counts below.

In response to this comment and others below, we calculated FPKM values for all genes in all samples and annotated the P/LP variants in **Supplementary Table 7A** with the FPKM values. The comprehensive expression analysis used to populate this table used GENCODE v19 as the source of gene models, as described in the **Methods**. In the original manuscript FPKM values on p.16 were computed using UCSC refFlat as the

gene model. We have corrected this discrepancy in the revised manuscript.

7. In the Table 6 summarizing “all pathologic findings”, the authors should include coverage/read support for SNVs and indels detected in PGCP/current study to help assess if sensitivity/specificity of detection is simply contingent upon coverage, not an intrinsic limitation of the sequencing platform.

[Response] We have added an additional **Supplementary Table, 7B**, with the read counts for SNVs and indels listed the original Supplementary table S6. Note that due to the addition of new supplementary tables to address reviewer feedback, the original Supplementary Table 6 is now **Supplementary Table 7A**.

Our response to the 4th point raised by the Reviewer #1 contains the additional analysis and **Supplementary Figure 3**, shown above, and clarifies that the variants missed by WES were caused by platform limitation rather than coverage along with a synopsis of variants detected only by WES.

8. Among “novel findings due to the inclusion of WGS”, the GLI2 amplification, TP53 indel or the DNMT3A deletion (Figure 6), or Notch 1-ITD (Supp Figure 7) do not strike one as inherently undetectable by WES or even targeted panels.

[Response] The focus of our SJMB030020 discussion was not GLI2 amplification - which is easily detected - but an intragenic TP53 deletion in the matched germline sample. Notably, this deletion was missed by Exome CNV in paired tumor/normal configuration and also by running the MB germline sample against an unrelated germline sample. This 22 kb deletion was below the resolution of Exome CNA prediction. Similarly, the 50 kb DNMT3 intragenic deletion was missed by Exome CNA. We clarified the unique contribution of WGS in the revision, adding comments to **p.16 paragraph 2** and to **p.17 paragraph 2**.

The new **Supplementary Figure 8**, created in response to the reviewer’s request for Exome CNV analysis, compares the CNAs missed by WES with those that were discovered by both platforms. The results show that CNAs affecting small and/or exon-poor regions would be challenging to detect using WES due to insufficient markers for robustly measuring read-depth changes. The two deletions (TP53 and DNMT3A) referred to by the reviewer belong to this category.

The NOTCH1-ITD was entirely absent from WES data. The ITD region was at the extreme edge of the captured region and coverage dropped off sharply into the intron. This is not surprising as the intronic regions are not targets for exome capture. We include comments to this effect on **p.17, paragraph 3**. We consider all three cases to be good, specific examples of the added value of WGS and are able to verify that the results remain valid even with the inclusion of Exome CNA analysis.

On the other hand, the failure to detect high level amplification of MYCN in a

retinoblastoma sample is quite intriguing. The fact that the clinical sample analyzed here was different from the PGC sample could be a plausible explanation, but is extenuated by the “almost perfect agreement” in the CNAs observed on chromosome 13 (Figure 5b).

[Response] Heterogeneity of the amplification between samples is the most likely explanation for the missing MYCN amplification. The focal high amplitude amplification in this sample implied an episomal amplification of MYCN. Since episome content is known to differ between cells within the same tumor, being well-described in medulloblastoma, for example (Ellison et al. 2011), we think this is a reasonable explanation for why the PGC sample contained the amplification but the sample in the present study did not. We have expanded on the above explanation in **Supplementary Note 4** entitled “Heterogeneity of MYCN Amplifications in SJRB051” included below:

The inconsistency between RB1 rearrangement and MYCN amplification implies that these events occurred at different times during tumor evolution. The DNA specimen for this study was extracted from a different vial from PGC (Supplementary Table 3, column K); however the bi-allelic RB1 re-arrangements are present in both specimens, suggesting that disruption of RB1 was an early event present in every tumor cell. In contrast, the MYCN amplification is likely to be a later event that is present in a subset of tumor cells. Given the high copy number and focal nature of the PGC sample’s MYCN amplification, episomal amplification via double minutes is a likely explanation (VanDevanter et al. 1990). Differing levels of oncogene amplification between cells of the same tumor has been previously reported in pediatric glioma (Paugh et al. 2011) and heterogeneity of MYC-bearing double-minutes has been previously reported in our study of medulloblastoma (Robinson et al. 2012).

9. The authors need to perform copy number analysis on WES data to make a fair comparison. They can perhaps try either “Falconx”, PureCN (PMC5157099) or Sequenza”, and see what works best. Though they cite an old reference (ref 37) in the discussion as a reason for not performing this analysis, several independent groups have been able to obtain this information from WES. Hence without these numbers from WES, the key data presented in Figure 5a (direct comparison between platforms) is highly misleading.

[Response]. We thank the reviewer for the suggestion on the methods for WES CNA analysis. We reviewed all three methods and selected Sequenza as it provided utilities to build reference genome mappability and GC window content. We also evaluated CNVkit, another recently published exome CNV algorithm that utilizes off-target reads to improve resolution (Talevich et al. 2016). Although we do not include the results from CNVkit, the comparison of Sequenza and CNVkit was very good; hence we felt comfortable using Sequenza as our Exome CNV algorithm. We have updated the text to include an explanation of our Sequenza pipeline used in contrast with our WGS copy number analysis including a comment in the Methods section on **p.29 paragraph 2** and throughout the “Detection of diverse types of pathogenic and likely pathogenic variants” section on **p.12-15**. We have also removed reference [37] from the manuscript. In the revised manuscript, **Fig. 5a** has been updated and now incorporates results from WES

CNV analysis (blue and orange segments) and ITDs (green segments) as shown below:

Figure 5 Detection of pathogenic or likely pathogenic variants by three platform sequencing. (a) Number of variants detected by each platform either alone or in combination.

10. The failure to detect FXR1-BRAF in a low grade glioma in the WGS data due to low tumor content is understandable, however, it is not mentioned if this fusion was not detected in RNA-seq data as well. That would be a bigger concern.

[Response] Neither WGS SV nor RNA-Seq SV detected the known FXR1-BRAF fusion. We added a comment to this effect on **p.13 paragraph 1**. We also added **Supplementary Note 6** containing further details:

Neither WGS SV nor RNA-Seq SV detected the known FXR1-BRAF fusion as documented in Supplementary Table 7A. Manual inspection of the aligned WGS data recovered two breakpoint reads, insufficient for an SV call by our pipeline. The average WGS coverage in the tumor was 36.8X (Supplementary Table 3) and the coverage at the two breakpoint regions was 54X and 36X. Manual inspection of aligned RNA-Seq reads at expected positions of exon fusion recovered one fusion read, also insufficient for the pipeline to call. The fusion was initially discovered by WGS in our research project, PCGP. The PCGP WGS coverage for this tumor was 65X and the BRAF fusion was detected by 3 junction reads and the estimated variant allele frequency (MAF) of 0.04 (Zhang et al. 2013). In RNA-Seq, FPKM for FXR1 and BRAF is 7.4 and 6.8 respectively. Therefore, we would conclude that the failure to identify the fusion in WGS and RNA-Seq was caused by the low MAF (0.04) of the fusion in a sample with low tumor purity.

Elsewhere, it is concerning that fusions including KMT2A-MLLT3, ETV6-RUNX1, KIAA1549-BRAF etc. were not detected. Can the authors provide data to support

the claim that this is due to low expression and not suboptimal yield of RNA-seq data?

[Response] The RNA-Seq coverage data presented in **Supplementary Table 4** shows SJLGG020, SJMLL019 and SJETV093 had 43-47% of the exons with >20X coverage with the average of the entire cohort as 47%. Therefore, the RNA-Seq coverage of all three samples was within the normal range for the study.

The FPKM values listed on **Supplementary Table 7A** show that KIAA1549 is expressed at a low level in SJLGG020 (FPKM of 4.31 and 10.01 for KIAA1549 and BRAF respectively). Low expression of KIAA1549, whose promoter drives the fusion, provides a good explanation for the absence of detectable KIAA1549-BRAF fusions transcripts in RNA-Seq, particularly as an estimated 70% of KIAA1549 expression would come from the wildtype haplotype based on estimated tumor purity from WGS data (**Supplementary Table 3**). Expression of KMT2A-MLLT3 in SJMLL019 (FPKM of 7.93 and 2.64 respectively) shows low expression of MLLT3. Even if all MLLT3 expression came from the fusion transcript, an FPKM of 2-3 implies low expression. Expression of ETV6-RUNX1 in SJETV093 (FPKM of 27.53 and 25.02 respectively) were not as low as the previous two examples, however RT-PCR showed that the fusion transcript itself was expressed at a low level. In summary, all of the fusions that RNA-Seq missed were expressed at low levels in tumor samples. We added a comment describing these findings on **p.14, paragraph 3** within the “Detection of diverse types of pathogenic and likely pathogenic variants”. These data as well as RT-PCR experiments are presented in **Supplementary Note 5**, entitled “**Additional Analysis on Gene Fusions Detected only by WGS**”:

We performed reverse transcriptase PCR (RT-PCR) to quantify expression levels of KMT2A-MLLT3 in SJMLL019 and ETV6-RUNX1 in SJETV093. Using standard procedures, we calculated ETV6-RUNX1 and KMT2A-MLLT3 expression relative to the house keeping gene, GAPDH in samples SJETV093 and SJMLL019 respectively as well as in positive control cell lines. The ETV6-RUNX1 fusion expression was approximately two logs lower than the level of GAPDH in both the control cell line and in SJETV093. KMT2A-MLLT3 showed a similar pattern with the expression of the fusion in the control cell line 1.5-2.0 logs lower than GAPDH. In SJMLL019, the expression of the fusion was approximately 5 logs lower than the expression of GAPDH (or 3 logs lower than the cell line fusion expression level). For reference, common fusions assayed on our lab including RUNX1-RUNXT1, TCF3-PBX1 and BCR-ABL are typically less than one log-fold lower than GAPDH. Cycle threshold (Ct) values are shown in **Supplementary Table 10**.

11. According to Supplemental Table S6 only two SNV/indels were not detected by WES (SH2B3 and FLT3). So in terms of the SNVs/Indels detection, WES performance is comparable to WGS in terms of LP/P events detection. The big difference then arises only because of amplification and deletions and SVs called from WGS, for which as stated above the numbers are not provided in this manuscript as CNV analysis are not done for WES.

[Response] There were 7 P/LP SNV/indels detected by WGS-only and 13 detected by WES-only (**Supplementary Table 7A**) because the variants marked “Rescue” in this Table had sufficient read count for cross-validation but not for de novo detection. We have clarified the meaning of this column in the table legend and in the **Methods** section on **p.26, paragraph 1** - Assessment of analytical performance. These seven variants have all been addressed above in our response to comment #4. We agree with the value of performing CNV analysis on WES, and we have addressed this issue in the preceding comments #3 and #9).

12. Sample SJMLL001 WES data has very high PCR duplication rate, probably due to sample quality issue, though the WGS library has good QC, and perhaps is the reason why FLT3 mutation was not detected by WES. Did the authors try repeat the WES library prep? What is the reason for low coverage in the SH2B3 indel sample (SJBALL021900)? Can the authors show the coverage in these regions?

[Response] The high WES duplication rate in SJMLL001 was due to sub-optimal libraries. Unfortunately, we were not able to repeat the library preparation for this case as input DNA was limited. Although the duplicate rate was high, the coverage statistics were calculated for the de-duplicated bam file and we clarify this in the revised **Methods** section of “Data Quality Control” on **p.27, paragraph 2**. Notably, the SJMLL001 WES tumor file had 100X coverage and was not an outlier as the mean WES coverage was 110X in this study (**Supplementary Table 3**).

Both the FLT3 and SH2B3 mutations were in regions that were poorly covered by WES across all 78 samples. We have added a **Supplementary Figure 3** (shown above in our response to point #4) detailing the coverage of the entire gene as well as at the variant site across the entire cohort. We have also added additional details to **Supplementary Tables 3 and 4** to show amounts of input DNA/RNA and more in-depth mapping statistics. Details about all 7 SNVs/indels missed by WES, including FLT3 and SH2B3, were also addressed in the response to the Reviewer’s comment #4 above.

13. The WES indel/SNV callers seem to give consistently higher number of calls than their WGS which is surprising (Suppl Figures 3 and 4). Is there a particular reason for this? Was the WES caller less stringent? This systemic discrepancy is probably reflected in the table in Suppl Figure 4b.

[Response] We applied the same methods using the same parameters for calling somatic SNVs and indels using WGS and WES, therefore the difference in the variant calling by WGS and WES is not due to the use of less stringent method for WES as asked by the Reviewer.

The number of somatic SNVs and indels presented in the original **Supplementary Figures 3 & 4** was based on variants in exons (i.e. e-SNVs and e-indels). To respond to the comment of this Reviewer and that of Reviewer 2, we have modified **Fig. 3** to incorporate the validation process and added **Supplementary Tables 6A-D** with statistics for both the SNVs and indels using the 18 samples that were also analyzed by WGS and

capture validation as part of the pediatric cancer genome project (PCGP). This will address the concern raised by Reviewer 2 that false positive and false negative rates were calculated based on different sets of samples.

We have added **Supplementary Note 1** entitled “SNV/indel call rate in WES” discussing the higher call rate of SNVs and indels in WES. In brief, the higher call rate in WES is driven by low-quality calls in low-complexity regions:

As shown in **Figure 3** and **Supplementary Tables 6A-D**, prior to quality filtering and cross-validation, there was a higher calling rate in WES than WGS for SNVs and more so for indels using the same threshold for variant detection (Zhang et al. 2012, Edmonson et al. 2011). Specifically, if we count only indels that had sufficient coverage in capture validation for ascertaining somatic mutation verification status as presented in **Supplementary Table 6B**, there are a total of 404 WES-only indels and 24 WGS-only indels. Of the 404 WES-only indels, only 14 (3.5% of 404) were of high quality while the vast majority (93.3%, 377 out of 404) were in highly repetitive regions of short tandem repeats (STR) or homopolymers of which nearly all (96.8%, 365/377) have low mutant allele fraction (MAF) of <0.1. Less than 4% of the WES-only indels were verified by custom capture. By contrast, of the 24 WGS-only indels, the majority (91.7%, 22/24) were of high quality and only a subset (33.3%, 8/24) were in repetitive regions. The majority (16 out of 24; 66.7%) of WGS-only indels were verified by custom capture even though many (15 out 24, 62.5%) had low MAF (<0.1).

The higher error rate of WES-only indels have been reported previously in a study that compared the indel genotype calls from WGS and WES in HapMap sample NA12878 (Fang et al. 2014). Our study confirmed the previous observation as the low validation rate (<4%) of those WES-only indels; the large majority of which had low MAF and were predominantly located in highly repetitive regions.

14. It appears that most notable SVs presented in the figures are all supported by fusion evidence from RNAseq. Here again while WGS does provide the evidence/details of the underlying genomic event, whether it adds to the calls among actionable/P/LP over RNAseq is not clearly presented?

[Response] We thank the reviewer for this comment and have worked to improve the presentation of the data. In 7 cases, only one platform detected a gene fusion with RNA-Seq missing three and WGS missing four. The three RNA-Seq misses are discussed above as part of the response to the Reviewer’s comment #10.

We had understated in the text the difficulty of predicting fusion genes from transcriptome data alone, especially for novel fusions. A recent survey of fusion prediction algorithms showed that even the best performing missed approximately 15% of known fusions but also made considerable numbers of additional predictions (Kumar et al. 2016).

When a fusion is well known, e.g. BCR-ABL1, it is easy to identify among a large number of false positive calls. However, fusions such as KDM6A-PTK2B, which

has not been well described, are much more difficult to call with certainty using RNA-Seq alone because typically many putative exon-exon fusions can be reported by an RNA-seq detection algorithm. In these cases, corroboration from DNA sequencing data with a copy number change point or SV junction in partner genes is required to ascertain the validity of a fusion transcript. Without such information, secondary validation becomes a necessary step adding to the cost and turnaround time.

Second, positional effects (i.e. enhancer hijacks) such as TCR-LMO2 and TCR-NKX2-1 in our samples are difficult to identify using RNA-Seq alone as they do not fuse disparate exons together but bring an enhancer into proximity of an oncogene. In these cases, having WGS support is required to differentiate cis-activation caused by translocations from trans-activation caused by aberrant expression in a regulatory gene. We added the following comment to **p.14 paragraph 3** within the “Detection of diverse types of pathogenic and likely pathogenic variants” section to make clear the additional yield of P/LP SVs found by a multiplatform approach.

For gene-fusing SVs, 83% (35/42) had two-platform support while 17% (7/42) were supported by only one platform. Specifically, WGS enabled detection of fusions with low expression (e.g. KMT2A-MLLT3, ETV6-RUNX1 and KIAA1549-BRAF) - which we confirmed by inspecting RNA-Seq expression values of partner genes (**Supplementary Table 7A**) and by performing RT-PCR (**Supplementary Note 5**) - whereas RNA-Seq recovered potentially repeat-associated and complex rearrangements difficult to detect or interpret using genomic DNA alone (e.g. RUNX1-RUNX1T1, BCL11A-GRIP2, FYCO1-RAF1 and KMT2A-AFF1). Thus, relying on RNA alone would have caused us to miss 3/42 gene fusions (7%). Additionally, WGS allowed us to unambiguously identify two T-cell receptor rearrangements, TCR-LMO2, with no RNA junction support but high expression and TCR-NKX2-1 with an RNA junction 1.3kb downstream of the target gene.

15. Why did the authors use older versions for several programs? For example, BWA current version is 7.2 and Picard current version is 2.92. Did the authors see any difference between these versions?

[Response] We used the BWA backtrack algorithm (aln/sampe) due to the length of our reads and because the CREST and Cicero algorithms require the soft-clip signature that is created by backtrack but not bwa mem. As seen in the bwa changelogs, the 0.6 and 0.7 lines have almost no changes to bwa aln. Furthermore, at the time the analysis was run, the 0.6 and 0.7 versions were not yet stable. We use Picard only for sorting, merging, and marking duplicates, and these straightforward functions have not changed significantly. We have added these explanatory remarks to the **Methods** on **p.28, paragraph 2**:

DNA reads were mapped using the backtrack algorithm (“aln” and “sampe” steps) of BWA 0.5.9 56. At the time the mapping was performed, BWA 0.5 was the latest stable version of BWA, as 0.6 and 0.7 were being released with frequent bug fixes. The changelogs of BWA versions 0.6 and 0.7 do not list any significant change to BWA backtrack. Aligned files were merged, sorted and de-duplicated using Picard tools 1.65 (broadinstitute.github.io/picard/).

16. From a per patient perspective, how many of the 78 cases benefit from WGS?

Can the authors discuss this in detail? This could very well be the basis of recommending a multi-tiered CLIA/sequencing approach where cost/time can be reduced for reporting. For example, WES (Tumor/normal) and RNAseq (Tumor) can be performed in the first round. Samples with more complex genome can be identified and recommended for WGS.

[Response] From **Figure 5** and **Supplementary Table 7A**, we show that including WGS as part of a three platform NGS test detected an additional 51 P/LP variants, 22% of the total P/LP variant count, otherwise stated as a 27% increase on the number of variants detected by a combination of Exome and RNA-Seq. On a per patient basis, 37/78 (47%) had one or more additional P/LP finding detected by the three platform test when compared to P/LP findings from Exome + RNA-Seq. We were cautious in reporting these numbers in the original manuscript as our pilot samples were somewhat pre-selected to contain known driving events. However, we agree that this is an important question and we add comments in the **Discussion** section on **p.18 paragraph 2** to address it.

We also note that this 22% figure is probably an over simplification. As 30% of SNVs and INDELS had a MAF of <20%. WES variant prediction in the 5-20% range continues to be challenging, and if only WES was used (or even with cross validation from RNASeq) there may still be a high burden of apparently high quality false positive calls.

A tiered approach might result in saving on sequencing cost; however, novel findings are likely to require validation and may not meet the turn-around-time required for real-time clinical practices. We have included this point in the discussion section on **p.21, paragraph 1** stating that:

As treatment of a pediatric cancer typically costs hundreds of thousands of dollars, the precision of integrative genomic profiling offered by three-platform sequencing outweighs the extended time for secondary validation and the potential for missing ~20% pathogenic variants in pediatric cancer.

Reviewer #2 (Remarks to the Author):

Rusch et al provide their experience of genomic profiling by total sequencing of whole-genome, whole exome, and whole transcriptome. The group has pioneered adoption of the advanced integrative next-generation sequencing to the field and there is inherent value in publishing their experience as more groups move along this path. There are several concerns of presentation that need to be addressed. The paper at times the paper reads as a branding or marketing presentation for a branded "Total Sequencing" which is unfortunate given the insight the results could provide and given total sequencing is a loaded phrase. The reporting of statistics was unclear and unfortunately seemed inconsistent. There was also insufficient discussion of the impact of sequencing depth on sensitivity, precision, and false negative rate given the low coverage sequencing typically expected for lower tumor content samples.

[Response] We thank the reviewer 2 for pointing out the value of genome-wide sequencing as an important direction for clinical genomics initiatives. In the revision, we replaced "**Total Sequencing**" with "**Three Platform Sequencing**", improved consistency in data presentation and enhanced our analysis and discussion on the impact of sequencing depth. The detailed responses are provided in each of the three areas pointed by the reviewer.

Major Concerns

Area 1. I found the branding of "Total Sequencing" to be unnecessary, ambiguous, and misleading given the phrase is highly loaded. It makes the paper ambiguous – should I review it on whether this is "total sequencing" or should I review the method "Total Sequencing"? I am choosing not to review on whether this is total sequencing, since it so clearly is not total sequencing, but rather optimized sequencing for deep exonic regions, mRNA, and relatively low pass cancer WGS. If the authors insist on it, could they more clearly (such as by a trademark symbol) distinguish this as branding? Even within regions that are well covered, triplicate expansions, inversions, and unphased DNA are just a couple of examples of categories that are missed. For example, there are several well-known repeat expansions that would be expected in these samples – the methods used do not manage expansions. The paper ends with Higher WGS coverage would further improve – which undermines total sequencing.

[Response] In the original manuscript, we used "Total" sequencing to refer to the 3-platform genome-wide sequencing employed in our study because it was concise and was named for the historic Total Therapy trials run at our institution. As the reviewer pointed that the use of the "Total" can be perceived as a branding, we replaced "Total" sequencing with "three platform" sequencing throughout the revised manuscript.

Area 2. I found the summary statistics very confusing and inconsistent, and at times

it seemed as selective reporting. Typically, sensitivity, specificity, and precision are provided in one table. These values are presented in a series of figures and tables. However, it seems the reportable range is changing. It is best to present these such that one can assess these values together, such as to avoid biases through selection of reportable range idea for each measure. For example, 4A appears to have 84 indels. Figure 3 (Precision) has 112 e-indel. I'm sure it's consistent, but this is some other examples have me wondering what the values are.

There needs to be more detail on the calculation of sensitivity, specificity, precision, FDR. Are the metrics cited in the abstract presented for other variant classes/types. For example, synonymous variants, intronic variants, transcription start-site variants, and many other classes that are critical part of discovery and these do not seem to be considered in the reported accuracy measurements. How were true negatives determined?

For example, in the cases where the tumor content was below 10% one would expect a large number of false negatives. Very specifically, the reportable range and the means by which true negatives were estimated need to be described. I wasn't sure for example if a manual rescue represents a true negative, false negative, or no result.

[Response] We acknowledge that the statistics presented in the previous manuscript were too complex. As described in the section titled "Sensitivity and Specificity" in the original manuscript, we calculated sensitivity based on 19 cases that were analyzed by various molecular pathology methods, our prior research program, PCGP, and the current study because it enabled us to identify false negatives. We calculated the positive predictive value (PPV, used interchangeably with specificity in the original manuscript) using all 38 samples that were subjected to capture validation because we wanted to calculate PPV by using a larger number of samples (38 instead of 19).

The reviewer pointed out that this approach could leave the impression of selective reporting. Therefore, in the revision, we decided to only include 18 PCGP-overlapping cases for reporting of the validation statistics. One of the original 19 cases was not included in this analysis because we found out during revision that capture validation was not performed in the original PCGP study. Focusing on these 18 cases allowed us to present all of the pertinent data in one place. This takes the form of a revised **Fig. 3** (shown below), an abridged version of **Fig. 4** and new **Supplementary Tables 6A-6D** that contain all of the variants we attempted to validate and their validation outcome. This simplification/consolidation led to removal of the original Supplementary Figs. 3 and 4 from the manuscript.

Figure 3. Sensitivity and positive predictive value of somatic variant detection based on capture validation of 18 cases with PCGP data. (a) Design of capture validation for measuring sensitivity and PPV of our analytical pipeline using somatic exonic SNV/indel detection as an example. (b) Summary of PPV for each variant type. Exonic indel and SNV are based on cross-validated results, whereas non-exonic SNV are based on WGS only; as such, they are reported separately. Our test does not report non-exonic indels, so they are omitted. (c) Summary of sensitivity for each variant type. For SNV/indel, most of the variants are detected by both platforms; of those that are only detected by one platform, results for WGS and WES were comparable, with slightly more detected by WGS.

In response to the reviewer’s comment on “estimation of true negatives”, we adopted a more precise term - “positive predictive value” (PPV) - which measures (number of true positive mutations)/(num. of true positive mutations + num. of false positive mutations) instead of “specificity”. We intentionally did not attempt to measure specificity because it would require evaluating the number of true/actual negatives across the entire exonic region, which can lead to deceptively high measures. Therefore, we replaced the term specificity with PPV throughout the revised manuscript and renamed the “Sensitivity and Specificity” section to “Analytical performance of somatic variant detection” on **page 10**.

For calculation of sensitivity, the false negatives are those that were verified by custom capture but were not detected or were filtered out by cross-validation in the clinical pilot study. This was clarified in the **Methods** section on **p.26** by including the following text:

Sensitivity and positive predictive value (PPV) were calculated using the set of variants that underwent capture validation and were covered after sequencing and alignment. We assessed PPV (true predicted positives divided by total predicted positives) rather than specificity (true predicted negatives divided by actual negatives) because the extremely high number of assessable positions inflates specificity beyond utility. For calculation of sensitivity, a variant was counted as a false negative in the clinical pilot study if it was not detected or was detected but filtered out by cross-validation.

The perception that we included samples with <10% tumor purity was caused by a confusing presentation in our original **Supplementary Table 3** as we put paired statistics

of purity/tumor-in-normal contamination in the same column. We have reformatted the table so that this and all other sample-specific information is in different columns for tumor and germline. It now clearly shows that all samples have tumor purity exceeding 20%. We clarified in the **Methods** section that “rescue” was a tag indicating the platform yielded sufficient evidence for cross-validation but not automated discovery. Therefore, when we discuss variants detectable by a single platform (e.g. WGS-alone or WES-alone), “rescue” is considered a false negative for that platform. We include the following clarification in the **Methods** section on **p.27, paragraph 1**:

In Supplementary Table 7A, variants that had support in a platform that was insufficient for detection but sufficient to cross-validate a call from another platform were labeled as Rescue. These variants are considered false negatives in the rescue platform alone.

To provide more details on the calculation of sensitivity and PPV, we modified **Fig. 3** to present the analytical and experimental process for calculating sensitivity and PPV based on capture validation results obtained from the 18 validation cases. The variant counts used for sensitivity and PPV calculation are now presented in a new **Supplementary Table 6D**. Although the conclusions remain unchanged, the precise number of variants varies slightly from the previous version due to the changes in the samples included for the analysis and manual correction of validation status of several variants. The revised manuscript also enlarges our description of the validation capture experiment as part of the “Hybrid Capture Validation Section” section of the **Methods** on **p.25-26**:

For Capture Validation (CapVal), we used the following algorithm to determine the validation status: First we obtained the mutant and reference allele counts for each marker in the corresponding sample, requiring a minimum base quality of 15 for SNVs and 5 for indels. A marker with 0 mutant reads and >20 total reads in tumor sample is called “Wildtype”. A Fisher’s Exact test was then performed on the tumor and normal CapVal read counts. When mutant reads are present in the normal samples, markers with Fisher’s Exact P value ≥ 0.05 or P value ≤ 0.05 and normal MAF ≥ 0.2 were called germline. Markers with P value ≤ 0.05 are called “Somatic”. To account for tumor-in-normal contamination (e.g. case SJHGG003_A), we required tumor MAF >0.05 if mutant reads were observed in the normal sample for “Somatic” calls. In addition, we manually inspected markers with P-value greater than 0.05 and adjusted the validation status to “SOMATIC” for 4 indels (annotated with an asterisk (*) in **Supplementary Table 6B**, due to reduced coverage in germline DNA or tumor-in-normal contamination (e.g. IFFO1.M364fs in SJHGG003_A). Since homopolymer artifacts are frequently observed in WES/CapVal, we also required ≥ 10 mutant reads in the CapVal data if the homopolymer marker was discovered by WES only, regardless of the CapVal depth.

As a result, 88 indels were called “SOMATIC” after validation, of which 3 were from PCGP. We identified 89 indels from our Clinical pipeline, of which 84 indels were covered in CapVal, and 83 validated as “SOMATIC”. This corresponds to a sensitivity of 94.3% (83/88) and a PPV of 98.8% (83/84). Similarly, 695 SNVs were called “SOMATIC” after validation, of which 34 were from PCGP. We identified 794 SNVs from our Clinical pipeline, of which 662 were covered in CapVal, and 653 were validated somatic. This corresponds to a sensitivity of 94% (653/695) and a PPV of 98.6% (653/662). For non-exonic SNVs (other SNVs), we are able to calculate PPV but not sensitivity because capture validation was performed only for variants detected and

passed the filters in the clinical pilot study.

The metrics cited in the abstract included all the variant classes (e.g. synonymous variants, intronic variants). However, we only placed them into the two broad categories - exonic variants and other (i.e. non-exonic) variants. This is because we can use WGS-WES for cross validation for variants in the exonic regions but not those in non-exonic regions. Not surprisingly the “other” SNV validation rate is slightly lower (97%) than the exonic SNVs (99%), so we put a range of validation rate (97-99%) in the revised abstract. We have also included **Supplementary Tables 6A-6C** in the revision - which lists each exonic SNV/indel (with their classifications such as silent, missense etc) and SV and associated validation status. Since we have ~13,000 other SNVs with capture validation data, we decided to host this data set online via St Jude Cloud instead of submitting it as a large supplementary table.

Area 3. Coverage is a critical aspect that is not quantitatively assessed in terms of a classic limits of detection analysis. The authors provide specificity and sensitivity, and discuss in great detail the importance of coverage. In practice for many clinical sequencing in CAP/CLIA environments the coverage they are sequencing is quite low and prone to false negatives. The various excel files highlight numerous examples where estimated tumor purity is less than 40%, less than 20%, and less than 10. The other concern is that the paper puts a major focus on whole-genome sequencing at relatively low coverage (30x) for mixed tumor/normal specimens, without going into the loss of sensitivity. Most clinical labs focusing on tumor sequencing often sequence to 400x -1,000x depth to examine lower tumor purity. The reported samples have a large portion of lower tumor content (50% are less than 50% tumor content), and so this discussion is relevant here. The authors should describe the dependence of sensitivity on tumor content of example.

[Response] The reviewer’s comment regarding many samples with very low tumor purity (less than 20% or 10%) is likely caused by an unclear presentation of tumor purity in **Supplementary Table 3**. In the original table, we had used column M to present the tumor purity for a tumor sample which has high percentage as well as tumor-in-normal contamination for a matching normal sample which has very low tumor content. To avoid this confusion, we now split the column M so that “tumor purity” and “tumor-in-normal contamination” are shown in two separate columns (I and Q) in the revised **Supplementary Table 3**. We also clarified in the main text on **p.6 paragraph 1** that the average tumor purity was 0.81 (range 0.21-1.00), and 14% (11 out of the 78) of the tumor specimens have purity <0.5. The “low purity” samples (i.e. <20% or <10% tumor content) referred to by the reviewer in the previous version of the manuscript are actually matching normal samples that had tumor-in-normal contamination.

To address the reviewer’s concern regarding the reduced power of 30X WGS for discovery of mutations with low mutant allele fraction (MAF), we included a power analysis in the revision (**Supplementary Fig. 4**). We compared the power for detecting somatic variants using the threshold implemented in our pipeline. The comparison

includes WGS-alone with 3-read, WGS+WES cross-validation (3 reads in WES + 1 WGS read and vice versa), and WES-alone with 10 reads for the variant allele (required for our pipeline as well as other published clinical pipelines for discovery/filtering for high-confidence variants in WES or gene panel) (Fang et al. 2014). We show that by using the cross-validation pipeline with 30X WGS and 100X WES, we are able to detect SNV/indels present with a MAF of 0.1 with >95% probability. By contrast 200X WES alone only gains <5% more power for detecting variants present at this MAF. 30X WGS-alone (requiring 3X mutant allele coverage) and 100X WES-alone (requiring 10X mutant allele coverage) only have 59% and 55% power for detecting variants at MAF <0.1. We did not go beyond 200X coverage in this analysis because our platform was genome-wide (WGS or exome) instead of a gene panel, and published WES coverage applied in a clinical setting is generally between 150 and 200X (Van Allen et al. 2014). A projected power for variant detection for 45X WGS was also included because this is the threshold used for our production clinical laboratory sequencing pipeline since the completion of the pilot study.

Supplementary Figure 4. Limit of detection analysis. The probability of detecting a variant with sufficient read evidence to make a call in WES and WGS is shown in black and green lines. We chose 100X WES and 30X WGS for our current study, 200X WES based on prior literature 61, and 45X WGS based on improvements made to our clinical genomics program after this pilot study completed. The blue lines show the probability of detection by at least 3 reads in WGS or WES and at least one read in the other platform, which is the standard used by this pipeline. Details of the calculations are in the

Supplementary Methods section.

We include an account of details of our limit of detection analysis in the **Supplementary Methods** section on **p.69-70** as presented below:

To assess the power to detect variants for our study design, we assumed a constant 100X for WES and 30X for WGS, and used binomial distribution to calculate the probability of observing ≥ 3 reads in one platform (as automatic detection) and observing ≥ 1 reads in the other platform (as being observed for validation purpose), for underlying MAFs ranged from 0.01 to 0.5, which correspond to cancer cell fraction of 2% to 100% in diploid regions. To calculate the probability of one variant being detected and validated, we multiply the probability of automatic detection in one platform (i.e., ≥ 3 mut reads) and the probability of observing the mutant reads in the other platform (i.e., ≥ 1 mut read), by assuming independence between samplings during WES and WGS sequencing. In addition, we performed re-sampling analysis to cross justify the above theoretical analysis, using NRAS G12D locus from case SJBALL021900 (with purity of 92%, and no sign of tumor in normal contamination, 53/110 in tumor WES, 19/45 in tumor WGS). For each predefined MAF, α , we sampled reads from tumor bam with probability $\alpha/0.92$ and from normal bam with probability $1-\alpha/0.92$ with replacement. 100 reads were sampled from tumor for WES and 30 reads from tumor for WGS.

In addition to the power calculation, we evaluated P/LP SNVs and indels that were detected by a single platform. Interestingly, 9 out of the 13 WES-specific variants (all cross-validated by WGS) were below 0.1 MAF, consistent with the expectation that 30X WGS alone has reduced power for detecting low MAF variants. By contrast, the majority of the 7 WGS-only variants have issues with WES capture. Of the 17 tumors with single-platform variants, 2 (12%) had $<50\%$ purity. Since only 11/78 tumors (14%) have purity lower than 50%, there is no enrichment for missing low MAF variants in low-purity samples in the current analysis. This shows that most of the low-MAF single-platform mutations were present in subclones of a tumor with high purity.

Reviewer #3 (Remarks to the Author):

The authors performed “total sequencing”, which includes whole genome (WGS), whole exome (WES) and whole transcriptome (WTS) sequencing, of 78 pediatric cancer patients. Their analysis pipeline detects, integrates, and cross-validates variants from each sequencing technique (platform). Experimental validation was performed to measure the accuracy of “total sequencing”. It claims that there is a critical need for using WGS in cancer screening in conjunction with WES and WTS.

The cross-validation method itself is not entirely novel, e.g. using WES to validate WGS (<https://www.ncbi.nlm.nih.gov/pubmed/22178993>) or using SV predictions to validate RNA fusions (<http://www.sciencedirect.com/science/article/pii/S0092867412010227>). The article maybe of interest to the clinical community to learn more about the utility of sequencing using multiple techniques. In the research community it is not an uncommon approach as cost is not a major concern but discovery unlike in a clinical setting that diagnostics have to be done routinely and have to be interpretable/actionable.

[Response] Cross-validation is a useful research tool and we are among the first to evaluate the feasibility and utility of cross validation in a CAP/CLIA environment. We do not wish to give the impression that cross-validation was our own invention, so we have added the suggested references to Lam et al. (2011) and Govindan et al. (2012) to the **Introduction on p.4, paragraph 3**. We have also explored value of cross validation as it pertains to limit of detection in response to Reviewer #2 and in the new **Supplementary Fig 4, shown above**. We have addressed concerns about cost as part of our response to Reviewer #1’s comments above.

It is commonly known to the community that the biggest limitation of WES and gene panel is on SV/CNV detection. There were also numerous suggestions to complement WES with WGS to fill that gap; e.g. using a low-cost, shallow WES for SV-driven tumors ([http://www.cell.com/trends/genetics/abstract/S0168-9525\(16\)30070-1](http://www.cell.com/trends/genetics/abstract/S0168-9525(16)30070-1)). On the other hand, WES or gene panel provides a more economical solution to sequence to very high depth (thousands of coverage) to detect low allele frequency variants in cancer, particularly in highly non-pure samples such as ctDNA. In that regard, Fig 4a in the article also showed that WES-only approach has higher sensitivity than WGS-only approach. Adding another sequencing can certainly validate the variants from WES; however, it is unclear in the article that how much more value it is for further validating detected, clinically actionable/interpretable variants which are usually having very high sensitivity and specificity in diagnostic assays.

[Response] The original Figure 4a (now **Figure 3c** - in response to feedback from Reviewer #2) showed sensitivity, with the bars divided into categories for those detected by both WGS and WES, only WGS, and only WES. The WGS-only approach actually

had a slightly higher sensitivity than WES-only, although difficult to see in the graph. We have clarified this in the figure legend that:

(c) Summary of sensitivity for each variant type. For SNV/indel, most of the variants are detected by both platforms; of those that are only detected by one platform, results for WGS and WES were comparable, with slightly more detected by WGS.

We have made all of the validation data available in new **Supplementary Tables 6A-C**. In addition, we would not advocate a WGS-only approach, as we feel that it is best to use WGS and WES in combination.

Regarding the ability of WGS to add value by detecting clinically actionable/interpretable variants, there are 7 pathogenic/likely pathogenic SNVs/indels found in this study that were not detected by WES, shown in **Supplementary Table 7A** (previously Supplementary Table 6). Of those 7, we would expect that 3-4 could not be recovered by WES alone with higher coverage based on manual inspection of the 7 variants, and based on adding additional WES reads from research sequencing of the same samples. We have updated the “Detection of diverse types of pathogenic and likely pathogenic variants” section in the main text **p.12, paragraph 2**, and added details in **Supplementary Note 2** entitled “Pathologic and likely pathologic mutations not detected by WES”, included below and in **Supplementary table 7C**.

A total of seven pathologic or likely pathologic somatic SNVs/indels were discovered by WGS alone. Of those seven two (FLT3 and SH2B3) had no support in WES, and five had insufficient support in WES for detection (**Supplementary Table 7C**). We manually inspected each of the seven to determine the cause for non-detection in WES. Four out of the seven sample also had whole exome sequencing from various research studies. For these four, we combined the reads from the clinical and research experiments and ran the resulting data through the analytical pipeline to determine if additional coverage would allow the variants to be detected by WES. The results are given in **Supplementary Table 7C**. In brief, three of the seven were in regions of systematically low coverage and would therefore be unlikely to be recovered by additional coverage (one had additional reads, and it was not detected with additional coverage). One appeared to be caused by poor capture of the indel-harboring fragments and would also be unlikely to be recovered by additional sequencing. The other three showed no signs of systematic WES-related problems and we all recovered with additional reads.

For other or rare variants that can only be detected by WGS, it is not clear how interpretable they are, as previously described e.g. in <https://www.ncbi.nlm.nih.gov/pmc/articles/PMC4484824/>.

We are conscious of the variant of uncertain significance problem. In addition to being detected, variants must also be interpreted. At present, this is a major challenge to the field. Regardless of the technology used to detect them, we need more sequenced samples to enable us to better interpret rare variants - especially in areas that are poorly captured in WES testing. In our study, we saw that combining data from multiple platforms, most

notably WGS and transcriptome actually helped us with the interpretation of variants by removing any ambiguity about their functional consequence. The PDGFRA, NOTCH1 and DNMT3A are used as illustrative examples of this.

References

1. Rennert, Eng, Zhang et al. (2016). Development and validation of a whole-exome sequencing test for simultaneous detection of point mutations, indels and copy-number alterations for precision cancer care. *NPJ Genom Med.* 1: 16019
2. Zehir, Benayed, Shah et al. (2017). Mutational landscape of metastatic cancer revealed from prospective clinical sequencing of 10,000 patients. *Nat Med.* 23(6):703-713
3. Favero, Joshi, Marquard et al. (2015). Sequenza: allele-specific copy number and mutation profiles from tumor sequencing data. *Ann Oncol.* 26(1):64-70
4. Chen, Gupta, Wang. *et al.* (2015). CONSERTING: integrating copy-number analysis with structural-variation detection. *Nat. Methods* 12, 527–530 (2015).
5. Zhang, Wu, Miller et al. (2013). Whole-genome sequencing identifies genetic alterations in pediatric low-grade gliomas. *Nat Genet.* 45(6):602-12
6. Dai, Kong, Si et al. (2013). Large-scale Analysis of PDGFRA Mutations in Melanomas and Evaluation of Their Sensitivity to Tyrosine Kinase Inhibitors Imatinib and Crenolanib. *Clin Cancer Res.* 19(24):6935-42
7. Ozawa, Brennan, Wang et al. (2010). PDGFRA gene rearrangements are frequent genetic events in PDGFRA-amplified glioblastomas. *Genes Dev.* 24(19):2205-18
8. Brennan, Verhaak, McKenna et al. (2013) The somatic genomic landscape of glioblastoma. *Cell* 155(2): 462–477.
9. VanDevanter, Piaskowski, Casper et al. (1990). *J Natl Cancer Inst.* 182(23):1815-21
10. Paugh, Broniscer, Qu et al. (2011). Genome-wide analyses identify recurrent amplifications of receptor tyrosine kinases and cell-cycle regulatory genes in diffuse intrinsic pontine glioma. *J Clin Oncol.* 29(30):3999-4006
11. Ellison, Kocack, Dalton, et al. (2011). Definition of disease-risk stratification groups in childhood medulloblastoma using combined clinical, pathologic, and molecular variables. *J Clin Oncol* 29(11): 1400-1407.
12. Robinson, Parker, Kranenburg et al. (2012). Novel mutations target distinct subgroups of medulloblastoma. *Nature* 488(7409):43-8
13. Talevich, Shain, Botton et al. (2016). CNVkit: Genome-Wide Copy Number Detection and Visualization from Targeted DNA Sequencing. *PLoS Comput Biol.* 12(4):e1004873

14. Fang, Wu, Narzisi et al. (2014) Reducing INDEL calling errors in whole genome and exome sequencing data. *Genome Med.* 6, 89
15. Kumar, Vo, Qin, Li et al. (2016). Comparative assessment of methods for the fusion transcripts detection from RNA-Seq data. *Sci Rep.* 6:21597
16. Van Allen, Wagle, Stojanov et al. (2014) Whole-exome sequencing and clinical interpretation of formalin-fixed, paraffin-embedded tumor samples to guide precision cancer medicine. *Nat Med.* 20(6):682-8
17. Lam, Clark, Chen et al. (2011). Performance comparison of whole-genome sequencing platforms. *Nat. Biotechnol.* 30(1): 78–82
18. Govindan, Ding, Griffith et al. (2012) Genomic landscape of non-small cell lung cancer in smokers and never-smokers. *Cell* 150(6): 1121–1134

Reviewers' Comments:

Reviewer #1:

Remarks to the Author:

The authors have provided a thorough response to reviewers' comments with many additional analyses and an excellent point-by-point response to questions. The manuscript should be acceptable for publication. I have no additional concerns.

Reviewer #2:

Remarks to the Author:

The resubmitted manuscript by Rushch et al addresses an important question, but largely fails to providing convincing evidence that is meaningful and leaves open considerable confusion about the statistics due to lack of clarity in the methods. The revised manuscript addresses points raised by the reviewers but there are concerns. Overall there wasn't sufficient evidence found for the primary conclusion: "The results of our study emphasize the critical need for incorporation of WGS in NGS-based screening approaches, particularly in the context of pediatric oncology." For example, WGS was done relatively low coverage and the benefit could be added ambiguity - how was this dealt with? Overall, the data just don't support the conclusion and lack of clarity of how statistics and final calls were obtained leave in doubt the validity of statistics. Specifically, it is unclear that the statistics consistently refer to the same set of calls. Sensitivity and PPV should be presented clearly within a table for each variant class for each assay separately. Then Sensitivity and PPV should be presented with the means of identifying the "Combined" calls. Manual calls should be clearly addressed so that we understand the extent of changes from automated methods. The methods were largely unclear and the reviewer was very uncertain how certain datasets were generated.

Detail

The use of manual calling brings into a lot of questions about scalability, and selective reporting of statistics. They need to provide greater detail on the times they over-ruled the algorithm produced calls or selected a call based on data that is different from the default algorithm.

One big concern is around statistics, and as presented they are confusing. By definition – PPV must be lower for the union of 3 non-identical variant sets that have some variants in common. Why is this not the case? Why are not the calls laid out by platform with their individual PPV's? The union will contain all the false positives from all methods – and thus false positives grow, while true positives remain the same. PPV increases with combined calls unless they are doing something more. If they are addressing this, it wasn't apparent in the method.

The title implies clinical sequencing – is this all from fixed pipelines within clinical tests or are these research calls within a CAP/CLIA lab. Figure 1 and much of the text bring various concerns – particularly that there are research calls being used mainly. How do they validate Chromothripsis calls? How do they evaluate limits of detection of ploidy? These just don't seem like assays that have undergone an analytical validation as a clinical test.

The paper still refers to total sequencing which the author acknowledges this is not.

Figure 3 is unclear and should be broken out by test such as within a table. Particularly Figure C is not helpful, and the caption does not make sense for Figure 3B

Supplementary Figure 4 and the limits of detection is done incorrectly and biased. They do not use the same calling methods as they do within the paper, instead referring to detection by at least 3 reads – which would lead to other false positive.

Reviewer #3:

Remarks to the Author:

The turnaround/cost discussion and the mentioning of the pathogenic and likely pathogenic variants discovered specifically in WGS are useful, and addressed my original concerns. Overall, the study analyzed a good number of patients in the cohort and demonstrated the value of a multi-platform sequencing approach with an integrative analysis that increase the precision of diagnosis. The study should be of interest to the community. Whether the extra value and the interpretability are worth the extra cost will be left to the community to judge, but at least translation research will benefit from the findings and approach.

Minor comments

- In line 1312, it says Four out of the seven sample, it should be "seven samples"
- Total sequencing is now changed to "three platform sequencing", which I believe should be "three-platform" sequencing
- On Fig 1a, it still says "total sequencing" instead of "three-platform sequencing", which should be consistent with the rest of the article.

Reviewers' comments:

Reviewer #1 (Remarks to the Author):

The authors have provided a through response to reviewers' comments with many additional analyses and an excellent point-by-point response to questions. The manuscript should be acceptable for publication. I have no additional concerns.

[Author Response] We are grateful for the reviewer's insightful comments during the first review which have led to major improvements in the current manuscript. We would like to thank the reviewer for acknowledging the importance of our study.

Reviewer #2 (Remarks to the Author):

The resubmitted manuscript by Rushch et al addresses an important question, but largely fails to providing convincing evidence that is meaningful and leaves open considerable confusion about the statistics due to lack of clarity in the methods. The revised manuscript addresses points raised by the reviewers but there are concerns. Overall there wasn't sufficient evidence found for the primary conclusion: "The results of our study emphasize the critical need for incorporation of WGS in NGS-based screening approaches, particularly in the context of pediatric oncology." For example, WGS was done relatively low coverage and the benefit could be added ambiguity - how was this dealt with? Overall, the data just don't support the conclusion and lack of clarity of how statistics and final calls were obtained leave in doubt the validity of statistics. Specifically, it is unclear that the statistics consistently refer to the same set of calls. Sensitivity and PPV should be presented clearly within a table for each variant class for each assay separately. Then Sensitivity and PPV should be presented with the means of identifying the "Combined" calls. Manual calls should be clearly addressed so that we understand the extent of changes from automated methods. The methods were largely unclear and the reviewer was very uncertain how certain datasets were generated.

[Author Response] To address this general comment, we have split it into four segments (a-d) separated by the gray highlight so that we can address each specifically. Changes to the manuscript in response to this reviewer are also highlighted in yellow in the revised manuscript.

[Reviewer 2 General comment a] *The resubmitted manuscript by Rushch [sic] et al addresses an important question, but largely fails to providing [sic] convincing evidence that is meaningful and leaves open considerable confusion about the statistics due to lack of clarity in the methods. The revised manuscript addresses points raised by the reviewers but there are concerns [sic].*

[Author Response] We have listed all somatic variants used for calculating statistics in Supplementary Table 6A-C and all P/LP variants in Supplementary Table S7A-B. To our knowledge, other published clinical genomics papers rarely publish all somatic variant data discovered in each case (Frampton et al, 2013; Roychowdhury et al, 2011; Parson et al, 2016; Harris et al 2016)—some only report summary data while others only report pathogenic variants. Additionally, we have taken the unusual step of making all data freely available on the cloud for those who wish to replicate our analysis. Therefore, not only is our analysis robust, we also provide the data at various level (including raw sequencing data, called somatic variants and summary data) to provide transparency and to enable reproducibility of our analysis. Examples will be given in our responses to other questions from this reviewer.

[Reviewer 2 General comment b] *Overall there wasn't sufficient evidence found for the primary conclusion: "The results of our study emphasize the critical need for incorporation of WGS in NGS-based screening approaches, particularly in the context of pediatric oncology." For example, WGS was done [at] relatively low coverage and the benefit could be added [sic] ambiguity - how was this dealt with? Overall,*

the data just don't support the conclusion and lack of clarity of how statistics and final calls were obtained leave in doubt the validity of statistics.

[Author Response] Our main conclusion, as stated in the discussion, is "As treatment of a pediatric cancer typically costs hundreds of thousands of dollars, the precision of integrative genomic profiling offered by three platform sequencing outweighs the extended time for secondary validation and the potential for missing ~20% pathogenic variants in pediatric cancer". The conclusion of 20% false-negative rate in reporting pathogenic variants by skipping WGS is based on the analysis presented in the section of **"Diagnostic Yield of two versus three platform sequencing"**. The reviewer is, presumably, agreeing to the 20% of missed variants as there is no critique raised regarding this number in the summary statement or in the specific comments. Therefore it is not clear the specific concerns that led to the reviewer's assessment on *"Overall, the data just don't support the conclusion"*.

Regarding the question *"For example, WGS was done relatively low coverage and the benefit could be added ambiguity - how was this dealt with?"*, first we want to clarify that the 30X WGS is by no means a relatively low coverage as low-pass WGS in clinical is defined as 5X to 15X (Roychowdhury et al. 2011). If the "ambiguity" in the reviewer's comment refers to potential false positives, our result presented in **Figure 3b** has shown that our variant calling, based on cross-validation between WGS and WES, is highly accurate. If the "ambiguity" refers to sensitivity, our power calculation presented at **Supplementary Fig. 4** has shown that by our current approach (30X WGS+100X WES), we are able to detect 95.7% of the variation with VAF ≥ 0.1 .

Our power calculation did show that by increasing WGS to 45X, we will increase the power for detection to 99.1%. For the ongoing clinical service work, we have implemented 45X WGS coverage (partly due to the reduced sequencing cost by HiSeq 4000), which was noted in the legend of **Supplementary Fig. 4**. To be more specific on the added benefit of increasing WGS from 30X to 45X, we included the following sentence in the revision in **Discussion**: "The increase in WGS coverage from 30x to 45x in clinical service, when integrated with 100x WES data, is expected to improve the power for detecting variants with variant allele frequency ≥ 0.1 from 95.7% to 99.1% (Supplementary Fig. 4)."

Regarding the comment on *"lack of clarity of how statistics and final calls were obtained leave in doubt the validity of statistics"*, the summary data presented in Supplementary Table S6D listed the count in each variant class and how validation statistics were calculated. Each count can be tracked to the matching variant record in Supplementary Table S6A-C. Details on variant calling were documented in the section of **"Mapping and SNV/indel Calling"** in **Methods**. To further improve clarity, we moved "variant filtering" from this section to a new section "SNV/Indel Filtering and Cross-validation" in the revision. This will provide a one-to-one mapping between the steps described in **Fig. 3, Methods**, and variant data presented in **Supplementary Tables S6**. More details are described in our response to **Reviewer 2 Specific comment 1**

[Reviewer 2 General comment c] Specifically, it is unclear that the statistics consistently refer to the same set of calls. Sensitivity and PPV should be presented clearly within a table for each variant class for each assay separately. Then Sensitivity and PPV should be presented with the means of identifying the "Combined" calls.

[Author Response] As we stated in the section **"Analytical performance of somatic variant detection"**, we used "18 cases that had been sequenced in the PCGP and had validation sequencing by custom capture" to measure sensitivity and PPV. Although the exact samples and their exact variants of these 18 cases are presented in **Supplementary Table S6A (for SNV), S6B (for indels) and S6C (for SVs)**, we realized that we never explicitly listed these samples in a single table. To improve clarity, we revised Supplementary Table S3 by including a new column "column B: inclusion for variant detection sensitivity and PPV analysis" to show the samples used for this analysis.

We respectfully disagree with the reviewer that "sensitivity and PPV should be presented within a table for each variant class for each assay separately". Integrating variant calls from WGS and WES by

the “cross-validation filtering” step outlined in **Fig. 3a** is what we implemented for clinical sequencing and also the main strength of our analytical approach. By using this approach, we were able to achieve high accuracy in variant calling sufficient to satisfy CLIA/CAP guidelines and two on site CAP inspections.

Presenting PPV and sensitivity for each assay (i.e. sequencing platform) without the integration process of “cross-validation filtering”, as recommended by the reviewer, would reflect the performance of published computation algorithms used in our workflow. These statistics were already presented in prior publications (Edmonson et al, 2011; Wang et al 2011; Chen et al 2015), therefore is out-of-scope for the present study and will be misleading as statistics generated from a single platform is incompatible with the multi-platform integrated analysis implemented in our study.

To further emphasize the importance of “cross-validation/filtering” in our study, we revised the section of “Mapping and SNV/indel Calling” in Methods by extracting the content related to cross-validation/filtering into a separate section “SNV/Indel Filtering and Cross-validation”. Given the importance of cross-validation in our analysis, we also expanded the description of cross-validation in this new section.

The summary statistics presented in Supplementary Table S6D are the final result generated from the “cross-validation and filtering” step that we implemented to integrate data from multiple platforms. This result was used as the baseline (i.e. predicted positives) for sensitivity and PPV analysis. In the revision, we further clarified that by modifying Figure 3 and by extending the Figure 3 legend. Details are presented in our response to specific questions below. These changes should further clarify that integrative analysis is not equivalent to “Combining” variant calls from multiple data sets.

Additionally, for any reader who is interested in obtaining summary statistics for a single platform, he or she can calculate the statistics based on the data in Supplementary Tables 6A-C where we record each variant, its discovery platform(s) by automated analysis, and filtering status based on our cross-validation filtering pipeline. For example, if a researcher is interested in knowing the sensitivity of exonic SNV detection on WGS platform alone, by restricting column A to “SOMATIC” and “SOMATIC*” and column B to “WGS_ONLY, WGS+WES” in Supplementary Table 6A, one could see that there are a total of 591 validated somatic variants by WGS. As the total number of true positive SNVs is 695 if we select all variants, we would be able to ascertain that the sensitivity for WGS platform is 591/695 (85%). To calculate the sensitivity of our integrative analysis pipeline for 3-platform sequencing, restricting column A to “SOMATIC” and “SOMATIC*” and selecting “PassFilter” in column C will give a total number of 653 which shows a sensitivity of 653/695 (94%). The column C is only relevant for 3-platform sequencing as the cross-validation filter utilizes multi-platform data. In the revision, the instructions for obtaining statistics for unfiltered variants is included in “Supplementary Note 7: Comparison of validation statistics on variants that passed the cross-validation filter with unfiltered variants”.

A segment of Supplementary Table S6A is shown below to illustrate this process.

A	B	C	D	E	F	G	H	I	J	K
CaptureVal Result	Clinical Sequencing Status		PCGP vs Clinical	Tumor	Chr	Pos	GeneName	mRNA_acc	Class	AAChange
	Platform	Pipeline								
WILDTYPE	WGS_ONLY	FailFilter	CLINICAL	SJEPD003_D	chr19	35790676	MAG	NM_002361	MISSENSE	G212V
SOMATIC	WGS_WES	PassFilter	BOTH	SJEPD003_D	chr6	38138729	BTBD9	NM_052893	UTR_3	E11_UTR_3
SOMATIC	WGS_WES	PassFilter	BOTH	SJEPD003_D	chr19	46717251	DKFZP434J0226	NR_027003	EXON	E6_EXON
SOMATIC	WGS_WES	PassFilter	CLINICAL	SJEPD003_D	chr11	65810554	GAL3ST3	NM_033036	SILENT	I240I
SOMATIC	WGS_ONLY	PassFilter	BOTH	SJEPD003_D	chr10	6627157	LOC439949	Unknown	EXON	E3_EXON
SOMATIC	WGS_WES	PassFilter	BOTH	SJEPD003_D	chr19	56347722	NLRP11	NM_145007	UTR_5	E1_UTR_5
SOMATIC	WGS_WES	PassFilter	BOTH	SJEPD003_D	chr19	56539089	NLRP5	NM_153447	MISSENSE	T497M
SOMATIC	WGS_WES	PassFilter	BOTH	SJEPD003_D	chr19	48558276	PLA2G4C	NM_003706	MISSENSE	R430C
SOMATIC	WGS_WES	PassFilter	BOTH	SJEPD003_D	chr1	205056124	RBBP5	NM_005057	UTR_3	E14_UTR_3
SOMATIC	WGS_WES	PassFilter	BOTH	SJEPD003_D	chr14	77844677	SAMD15	NM_001010860	MISSENSE	E306Q
SOMATIC	WGS_WES	PassFilter	BOTH	SJEPD003_D	chr1	10238845	UBE4B	NM_001105562	SILENT	I1223I

[Reviewer 2 General comment d] Manual calls should be clearly addressed so that we understand the extent of changes from automated methods. The methods were largely unclear and the reviewer was very uncertain how certain datasets were generated.

[Author Response] We do NOT manually call variants. The new section of “**SNV/Indel filtering and cross-validation**” in **Methods** should clarify this point. The details are describe in our response to “**Reviewer 2 Specific comment 1**”.

[Reviewer 2 Specific comment 1] The use of manual calling brings into a lot of questions about scalability, and selective reporting of statistics. They need to provide greater detail on the times they over-ruled the algorithm produced calls or selected a call based on data that is different from the default algorithm.

[Author Response] We did not manually call variants—all variants were called and filtered by automated pipelines as described in the section of “**Mapping and SNV/indel Calling**” as well as in the new section of “**SNV/Indel Filtering and Cross-validation**” in **Methods**. Manual review was used to verify exonic mutations (SNVs and indels) but NOT to make variant calls. For non-exonic mutations, we did not perform manual review as these variants are not reportable in the current clinical sequencing program.

Furthermore, we want to emphasize that manual inspection of called variants is a standard practice in molecular pathology labs. For example, a recent publication by MSKCC on Tumor molecular profiling (Zehir et al, *Nat Med.* 2017 Jun;23(6):703-713) also described the need for manual review as follows: “DNA isolated from tumor tissue and, in 98% of cases, matched normal peripheral blood was subjected to hybridization capture and deep-coverage NGS to detect somatic mutations, small insertions and deletions, CNAs and chromosomal rearrangements, **all of which were manually reviewed**”.

We did not record how often a variant call, which passed the automated computational process involving Bambino call, quality filtering and cross-validation, was rejected by manual review. However, the 2% difference on the validation rate between exonic SNVs (99%) which involves manual review and non-exonic SNVs (97%) which relies on the automated process alone suggest that manual review may have improved PPV by 2%.

In the revised manuscript we described the manual review process in the new section of “**SNV/Indel Filtering and Cross-validation**” in **Methods** as follows: “Manual review was performed for exonic variants that passed the above mentioned computation filtering. Non-exonic variants were not manually reviewed. Given the 2% difference on the validation rate between the manually-reviewed exonic SNVs (99%) and automatically computed non-exonic SNVs (97%), we estimate that manual review improves PPV by 2%.”

Throughout the manuscript we report the statistics on variants that were computed from our analytical pipelines which include the cross-validation filtering. We respectfully disagree with the reviewer that this is “selective reporting”.

[Reviewer 2 specific comment 2] One big concern is around statistics, and as presented they are confusing. By definition – PPV must be lower for the union of 3 non-identical variant sets that have some variants in column. Why is this not the case? Why are not the calls laid out by platform with their individual PPV’s? The union will contain all the false positives from all methods – and thus false positives grow, while true positives remain the same. PPV increases with combined calls unless they are doing something more. If they are addressing this, it wasn't apprent [sic] in the method.

[Author Response] As outlined in our response to **[Reviewer 2 General comment c]**, variants presented to our clinical analysts are those that passed the “cross-validation filtering” implemented as part of our clinical sequencing analysis pipeline. Therefore, use of filtered variants for calculate PPV is appropriate as it matches to the real implementation of our clinical pipeline. We have documented this

in the manuscript as follows: *“To determine positives for calculating PPV, we used final variant calls, which were the product of quality filtering, cross-validation analysis, and manual review for all mutation types except for non-exonic SNVs” in section of “Analytical performance of somatic variant detection”.*

Presenting PPV of a single-platform (requested by reviewer 2 in the general comment) or that of the union of WGS/WES would be inappropriate as our study did NOT take the union of variants generated from WGS/WES and assess their clinical relevance. *In the revision, we further clarified that only WGS and WES were used for SNV/indel detection in the section of “Mapping and SNV/indel Calling” in Methods while RNA-seq was used for cross-validation filter of SNV/indel as outlined in the new section of “SNV/Indel Filtering and Cross-validation”.* *This new section should also clarify that the final variants are not the union of WGS and WES but variants cross-validated by WGS and WES.*

In the design of experimental validation by capture sequencing we did include the union of WGS, WES and those previously validated in our research project PCGP to develop a “truth” data set for assessing sensitivity of our clinical pipeline as shown in **Figure 3a**. Some of the real somatic variants detected by a single platform could have been filtered by the “cross-validation filtering” and they were counted as false negatives for our clinical pipeline. For example, in **Supplementary Table 6A**, there are a total of 8 somatic exonic SNVs that were detected by a single platform (WGS or WES) but did not pass “cross-validation filter” and were thus considered false negatives in our calculation of sensitivity. The truth data set also included an additional 34 SNVs that were detected only in our research project PCGP. This design was presented in **Figure 3a** and details on the number used for calculating PPV and sensitivity are described in the section of **“Hybrid Capture Validation” in Methods**.

As described in our response to **[Reviewer 2 General comment c]**, a curious reader can easily obtain PPV for a single platform or the union of WGS/WES by using the data on **Supplementary Tables 6A-C**. We do feel strongly that the PPV should be based only on the filtered results in the manuscript as it reflects the strength of our cross-platform integrative analysis. Therefore, we choose not to present any statistics in the single-platform or in the union of WGS/WES without cross-validation filtering. *In the revised manuscript we included instructions on how to obtain statistics of unfiltered variants for readers who are interested in such statistics. The instruction is included in the section of “Comparison of validation statistics on variants that pass cross-validation filter with unfiltered variants” in Supplementary Note 7.* The example included in Supplementary Note 7 should allow the reviewer to see the difference between the statistics calculated from unfiltered variants versus those that pass cross-validation filter.

[Reviewer 2 specific comment 3] *The title implies clinical sequencing – is this all from fixed pipelines within clinical tests or are these research calls within a CAP/CLIA lab. Figure 1 and much of the text bring various concerns – particularly that there are research calls being used mainly. How do they validate Chromothripsis [sic] calls? How do they evaluate limits of detection of ploidy? These just don’t seem like assays that have undergone an analytical validation as a clinical test.*

[Author Response] The entire workflow is part of the clinical validation of the testing which includes wet bench through the determination of the final calls and patient report. At the conclusion of that work, the entire process was written up as the validation which was reviewed on the next CAP inspection that we had after the validation was signed off on. The CAP inspector felt that the work and analysis performed met all the requirements of CAP at that time. At that point, it became a CAP/CLIA validated test. This was also done prior to the current CAP requirements as this work was completed prior to the current guidelines. At the time the work was performed, we had far surpassed any requirements in the CAP checklist. *The WES CNV analysis was not part of the clinical testing as we incorporated this analysis for comparison with WGS CNV analysis. We clarified that in the revised manuscript in the section of “SV/CNV/Fusion Calling” in Methods.*

The ploidy test was included for CAP inspection through the comparison with data generated by cytogenetic lab and there are a total of 9 such events documented in Supplementary Table S2. Chromothripsis was used as an annotation for ploidy report and we followed guidelines by Korbel and Campbell (2013). We have clarified this in the revised legend of Figure 1.

[Reviewer 2 specific comment 4] *The paper still refers to total sequencing which the author acknowledges this is not.*

[Author Response] We changed the term “Total Sequencing” to three-platform sequencing throughout the manuscript. As reviewer 3 points out, we missed a single instance in the body Figure 1a. This was an oversight on our part and has been corrected in the revision.

[Reviewer 2 specific comment 5] Figure 3 is unclear and should be broken out by test such as within a table. Particularly Figure C is not helpful, and the caption does not make sense for Figure 3B

[Author Response] We assume that the reviewer is requesting to have PPV/sensitivity be measured by each sequencing platform (i.e. “by test”) based on prior comments. We have presented the rationale for presenting PPV and sensitivity using the variants that pass the cross-validation filter in our response to **Reviewer 2 General comment c**, i.e. our analysis pipeline is based on integrating WGS and WES data for cross-validation instead of making a union of variant calls from the two platforms. As described in previous response, we included instruction on how to calculate statistics for unfiltered variants in a new section of **“Supplementary Note 7: Comparison of validation statistics on variants that pass cross-validation filter with unfiltered variant”**.

We have described the use of “cross-validation filtered” variants in the analysis of PPV in the section of “Analytical performance of somatic variant detection”. In the revision we expanded the legend of Figure 3a to further clarify the filtered variants are considered “predicted positive”. We also provided additional details on the need to separate non-exonic (other) SNV from exonic SNV as a separate category in the revised legend of Figure 3b. The added content in the legend, although generally a repetition of what we presented in the main text in the section of “Analytical performance of somatic variant detection”, should improve the clarity of Figure 3.

Figure 3c aims to depict the sensitivity of 3-platform sequencing, which is important for clinical assay, therefore we respectfully disagree with the reviewer that it is “not helpful”. The sensitivity of the assay is a critical result of the study and we invested tremendous amount of effort in capture validating all the raw variant calls (including those that were filtered out or not detected in our pipeline so that we can ascertain the sensitivity. We maintain that it is important for sensitivity to appear in a main figure. Based on the reviewer’s comment, we recognize that categorizing validated variants by WGS, WES or PCGP in Figure 3a may cause confusion and we removed this component in Figure 3a. Instead, we added description of “WGS+WES”, “WGS”, “WES” and “Missed” to the legend of Figure 3c.

An in-depth explanation of the methods and data behind the figure is given in the Results, Methods, supplementary tables, and the newly added Supplementary Note 7. Although the full content cannot be presented in a figure caption, we have revised Figure 3 and the caption to improve clarity. We maintain that it is important for sensitivity of three-platform sequencing to appear in a main figure.

[Reviewer 2 specific comment 6] Supplementary Figure 4 and the limits of detection is done incorrectly and biased. They do not use the same calling methods as they do within the paper, instead referring to detection by at least 3 reads – which would lead to other false positive.

[Author Response] Figure S4 legend reads “The blue lines show the probability of detection by at least 3 reads in WGS or WES and at least one read in the other platform, which is the standard used by this pipeline.” This simulation is based on the same criteria that we used for automated call defined in **Methods** “The variant allele required a minimum of three supporting reads” and the minimum of 1

read from another platform for cross validation as defined in the section of “**SNP/Indel filtering and cross-validation**”. In the revision, we expanded the Figure S4 legend by changing “which is the standard used by this pipeline” to “which is the standard used by the automated detection and cross-validation filtering pipeline” to improve clarity.

Figure S4 was generated from a theoretical analysis; and we were able to perform a re-sampling analysis to justify the above theoretical analysis using *NRAS* G12D locus from case SJBALL021900. This was presented in the section of “**Limit of Detection Analysis**” in **Supplementary Methods**.

Reviewer #3 (Remarks to the Author):

The turnaround/cost discussion and the mentioning of the pathogenic and likely pathogenic variants discovered specifically in WGS are useful, and addressed my original concerns. Overall, the study analyzed a good number of patients in the cohort and demonstrated the value of a multi-platform sequencing approach with an integrative analysis that increase the precision of diagnosis. The study should be of interest to the community. Whether the extra value and the interpretability are worth the extra cost will be left to the community to judge, but at least translation research will benefit from the findings and approach.

[Author Response] We thank the reviewer for his/her understanding of the importance of our study which involves integrative analysis of multi-platform sequencing in a clinical setting.

Minor comments

- *In line 1312, it says Four out of the seven sample, it should be "seven samples"*

[Author Response] We made the correction to “seven samples” in the revised manuscript.

- *Total sequencing is now changed to "three platform sequencing", which I believe should be "three-platform" sequencing*

[Author Response] We made the correction throughout the text to “three-platform”.

- *On Fig 1a, it still says "total sequencing" instead of "three-platform sequencing", which should be consistent with the rest of the article.*

[Author Response] We thank the reviewer for pointing out this omission and have revised Fig. 1a to convert “total sequencing” to “three-platform sequencing”.

References

Frampton, G. M. et al. Development and validation of a clinical cancer genomic profiling test based on massively parallel DNA sequencing. *Nat. Biotechnol.* 31, 1023–1031 (2013).

Roychowdhury, S. et al. Personalized oncology through integrative high-throughput sequencing: a pilot study. *Sci. Transl. Med.* 3, 111ra121 (2011).

Parsons, D. W. et al. Diagnostic Yield of Clinical Tumor and Germline Whole-Exome Sequencing for Children With Solid Tumors. *JAMA Oncol.* 2, 616–624 (2016).

Harris, M. H. et al. Multicenter Feasibility Study of Tumor Molecular Profiling to Inform Therapeutic Decisions in Advanced Pediatric Solid Tumors: The Individualized Cancer Therapy (iCat) Study. *JAMA Oncol.* 2, 608–615 (2016).

Zehir, A et al. Mutational landscape of metastatic cancer revealed from prospective clinical sequencing of 10,000 patients. *Nat Med.* 23:703-713 (2017)

Korbel, J. O. & Campbell, P. J. Criteria for inference of chromothripsis in cancer genomes. *Cell* 152, 1226–1236 (2013).

Reviewers' Comments:

Reviewer #2:

Remarks to the Author:

The manuscript is much improved. No additional comments.